# KLHL41 stabilizes skeletal muscle sarcomeres by nonproteolytic ubiquitination

Andres Ramirez-Martinez[1,2,3], Bercin Kutluk Cenik[1,2,3], Svetlana Bezprozvannaya[1,2,3], Beibei Chen[4], Rhonda Bassel-Duby[1,2,3], Ning Liu[1,2,3]*, Eric N Olson[1,2,3]*

[1]Department of Molecular Biology, University of Texas Southwestern Medical Center, Dallas, United States; [2]Hamon Center for Regenerative Science and Medicine, University of Texas Southwestern Medical Center, Dallas, United States; [3]Senator Paul D. Wellstone Muscular Dystrophy Cooperative Research Center, University of Texas Southwestern Medical Center, Dallas, United States; [4]Department of Clinical Sciences, University of Texas Southwestern Medical Center, Dallas, United States

*For correspondence:
Ning.Liu@utsouthwestern.edu
(NL);
Eric.Olson@UTSouthwestern.edu
(ENO)

**Competing interests:** The authors declare that no competing interests exist.

**Abstract** Maintenance of muscle function requires assembly of contractile proteins into highly organized sarcomeres. Mutations in Kelch-like protein 41 (KLHL41) cause nemaline myopathy, a fatal muscle disorder associated with sarcomere disarray. We generated KLHL41 mutant mice, which display lethal disruption of sarcomeres and aberrant expression of muscle structural and contractile proteins, mimicking the hallmarks of the human disease. We show that KLHL41 is poly-ubiquitinated and acts, at least in part, by preventing aggregation and degradation of Nebulin, an essential component of the sarcomere. Furthermore, inhibition of KLHL41 poly-ubiquitination prevents its stabilization of nebulin, suggesting a unique role for ubiquitination in protein stabilization. These findings provide new insights into the molecular etiology of nemaline myopathy and reveal a mechanism whereby KLHL41 stabilizes sarcomeres and maintains muscle function by acting as a molecular chaperone. Similar mechanisms for protein stabilization likely contribute to the actions of other Kelch proteins.
DOI: https://doi.org/10.7554/eLife.26439.001

## Introduction

Nemaline myopathy (NM) is one of the most severe forms of congenital myopathy, affecting ~1 in 50,000 births (*Romero et al., 2013*; *Wallgren-Pettersson et al., 2011*). NM encompasses a set of genetically heterogeneous diseases defined by the presence of rod-like structures, called nemaline bodies, in skeletal muscle fibers. Nemaline bodies are formed by the abnormal aggregation of proteins within muscle thin filaments and are associated with myofibril disorganization, reduced contractile force, mitochondrial dysfunction, and a spectrum of abnormalities ranging from mild muscle weakness to complete akinesia. There is no effective treatment for NM and mechanistic insight into the molecular basis of the disorder is lacking.

To date, mutations in 11 different genes have been linked to NM, most of which encode components of the sarcomeric thin filament, including nebulin (NEB), actin-1, tropomyosins-2 and -3, troponin-1 and leiomodin-3 (LMOD3) (*Agrawal et al., 2004*; *Cenik et al., 2015*; *Donner et al., 2002*; *Johnston et al., 2000*; *Lehtokari et al., 2006*; *Wattanasirichaigoon et al., 2002*; *Yuen et al., 2014*). Interestingly, 3 genes associated with NM, *KLHL40*, *KLHL41* and *KBTBD13*, encode Kelch proteins, characterized by the presence of a Kelch-repeat domain, a BTB/POZ domain involved in

**eLife digest** Together with the tendon and joints, muscles move our bones by contracting and relaxing. Muscles are formed of bundles of lengthy cells, which are made up of small units called sarcomeres. To contract, the proteins in the sarcomere need to be able to slide past each other. In healthy muscle cells, the proteins in the sarcomeres are evenly distributed in an organized pattern to make sure that the muscle can contract.

However, when some of the proteins in the sarcomere become faulty, it can lead to diseases that affect the muscles and movement. For example, in a genetic muscle disease called nemaline myopathy, the proteins in the sarcomere are no longer organized, which leads to a build-up of proteins in the muscle fiber. These protein masses form rod-like structures that are lodged in between sarcomeres, which makes it more difficult for the muscles to contract. This can cause muscle weakness, difficulties eating or breathing, and eventually death.

A common cause of nemaline myopathy is mutations in the gene that encodes the nebulin protein, which serves as a scaffold for the sarcomere assembly. Other proteins, including proteins named KLHL40 and KLHL41, have also been linked to the disease. These two proteins are both 'Kelch' proteins, most of which help to degrade specific proteins. However, a recent study has shown that KLHL40 actually stabilizes nebulin. As the two Kelch proteins KLHL40 and 41 are very similar in structure, scientists wanted to find out if KLHL41 plays a similar role to KLHL40.

Now, Ramirez-Martinez et al. have created genetically modified mice that lacked KLHL41. These mice showed symptoms similar to people with nemaline myopathy, in which the sarcomeres were disorganized and could not form properly. Further experiments with cells grown in the laboratory showed that KLHL41 stabilized nebulin by using a specific chemical process that usually helps to degrade proteins.

These results suggest that Kelch proteins have additional roles beyond degrading proteins and that some proteins linked to nemaline myopathy may actively prevent others from accumulating. A next step will be to find drugs that can compensate for the lack of KLHL41. A better understanding of the causes of this fatal disease will contribute towards developing better treatments.
DOI: https://doi.org/10.7554/eLife.26439.002

protein-protein interaction, and a BACK domain that binds E3 ubiquitin ligases (*Geyer et al., 2003*; *Gupta and Beggs, 2014*). More than 60 different Kelch proteins have been identified (*Dhanoa et al., 2013*), many of which function as substrate-specific adaptors for Cullin-3 (CUL3) E3 ubiquitin ligase, a component of the ubiquitin-proteasome system (*Genschik et al., 2013*). Modification of proteins by the covalent attachment of ubiquitin to lysine residues via CUL3 serves as a signal for protein degradation. However, there are also Kelch proteins that function independently of proteolysis (*Werner et al., 2015*) and much remains to be learned about their functions.

Recently, we and others showed that loss of function of the muscle-specific Kelch protein KLHL40 in mice causes NM similar to that seen in human patients with KLHL40 mutations (*Garg et al., 2014*; *Ravenscroft et al., 2013*). Unlike other Kelch proteins that promote protein degradation, KLHL40 is required for stabilization of LMOD3, an actin nucleator, and NEB, a molecular ruler that controls myofibrillogenesis (*Cenik et al., 2015*; *Garg et al., 2014*). The absence of LMOD3 or NEB causes lethal NM and severe disruption of skeletal muscle sarcomeric structure and function in mice and humans, confirming the essential roles of these proteins in maintenance of sarcomere integrity. KLHL40 shares 52% identity with KLHL41, which is also expressed in a muscle-specific manner (*Taylor et al., 1998*). Similarly, KLHL41 mutations in humans have been associated with NM, and morpholino knockdown of KLHL41 in zebrafish causes NM-like abnormalities with aberrant myofibril formation (*Gupta et al., 2013*). However, the molecular functions of KLHL41 and the mechanistic basis of these abnormalities have not been determined.

Here, we describe a mouse model of severe NM caused by a loss of function mutation in the *Klhl41* gene. Although KLHL40 and 41 share high homology and muscle-specific expression, we show that their mechanisms of action are distinct. KLHL41 preferentially stabilizes NEB rather than LMOD3, and this activity is dependent on poly-ubiquitination sensed through the BTB domain of KLHL41. In the absence of KLHL41, NEB and other sarcomeric components fail to accumulate,

resulting in early neonatal lethality. These findings provide new insight into the molecular etiology of NM and also reveal a previously unrecognized role for Kelch proteins in protein stabilization and chaperone activity.

## Results

### Muscle-specific expression of *Klhl41*

*Klhl41* is expressed in a muscle-specific manner with highest levels in adult skeletal muscle relative to the heart (*Figure 1A–B*). During embryogenesis, *Klhl41* is highly expressed in somite myotomes at embryonic day (E) 10.5 and in skeletal muscles throughout the body at later stages, as detected by in situ hybridization (*Figure 1C*).

### *Klhl41* knockout mice display neonatal lethality

To investigate the function of *Klhl41* in mice, we obtained embryonic stem cells that contained a LacZ-Neo cassette in intron 1 of the *Klhl41* locus from KOMP (*Figure 1—figure supplement 1*). In this allele, exon 1 of the *Klhl41* gene is predicted to be spliced to LacZ, preventing expression of a functional *Klhl41* transcript. Mice heterozygous for the mutant allele ($Klhl41^{+/-}$) were normal and were intercrossed to obtain $Klhl41^{-/-}$ knockout (KO) mice. LacZ expression was not detected in $Klhl41^{+/-}$ mice, suggesting that the LacZ cassette was spliced out. Quantitative RT-PCR (qRT-PCR) confirmed the complete loss of *Klhl41* transcript in KO mice (*Figure 2A*) and western blot analysis revealed loss of KLHL41 protein in muscle from KO mice (*Figure 2B*). KO mice were born at Mendelian ratios from heterozygous intercrosses and were indistinguishable from WT littermates at birth (*Figure 2C*). However, KO pups failed to thrive and showed progressive lethality from birth to post-natal day (P) 12, after which no surviving KO mice were observed (*Figure 2D*). In order to ascertain that the failure to thrive was not due to difficulty in suckling or breathing, we confirmed that KO mice had milk spots comparable to those of their WT littermates. KO mice that survived the early neonatal period displayed severe runting at P3 and P10 (*Figure 2C*), and their body weight failed to increase with age (*Figure 2E*), even when other littermates were removed from their mothers, indicating an intrinsic abnormality rather than simply an inability to compete with stronger siblings for nursing.

### Loss of KLHL41 leads to nemaline myopathy

Histological analysis of skeletal muscles of KO mice at various time points revealed occasional ragged fibers (fibers with discontinuities in their staining) in the diaphragm and the hindlimbs at P0 and P10 (*Figure 3A* and *Figure 3—figure supplement 1A*). Further abnormalities in muscle histology were observed by Gomori's trichrome staining (*Figure 3B*). KO myofibers presented abundant depositions that were absent in WT muscle. Desmin staining showed additional cytoskeletal disarray (*Figure 3C*). In WT muscle at P3, desmin was evenly expressed throughout transverse sections of myofibers. However, in KO mice, desmin protein aggregates were distributed across myofibers, indicating abnormal sarcomere structure (*Figure 3C*). We observed a similar general disarray in sarcomeric α-actinin staining in KO muscle (*Figure 3—figure supplement 1B*).

Electron microscopy of diaphragm (*Figure 3D–E*) and hindlimb muscle (*Figure 3—figure supplement 2*) at P3 also showed sarcomere disarray and Z-line streaming, as well as electron dense inclusions, corresponding to nemaline bodies. Hearts from WT and KO mice were indistinguishable (*Figure 3—figure supplement 1A*).

### KLHL41 is required to maintain normal levels of sarcomeric proteins in vivo

To further define the muscle abnormalities of *Klhl41* KO mice, we compared the protein compositions of WT and *Klhl41* KO hindlimb muscle at P0 by unbiased quantitative proteomics. These studies revealed a total of 389 proteins that were up- or down-regulated in muscle from the KO mice, with KLHL41 being the most down-regulated protein (*Figure 4A* and *Figure 4—source data 1*). Analysis of enriched biological pathways using the Database for Annotation, Visualization and Integrated Discovery (DAVID) (*Jiao et al., 2012*) revealed that 'sarcomere organization' and 'regulation of muscle contraction' proteins were aberrantly down-regulated in KO mice (*Figure 4—figure*

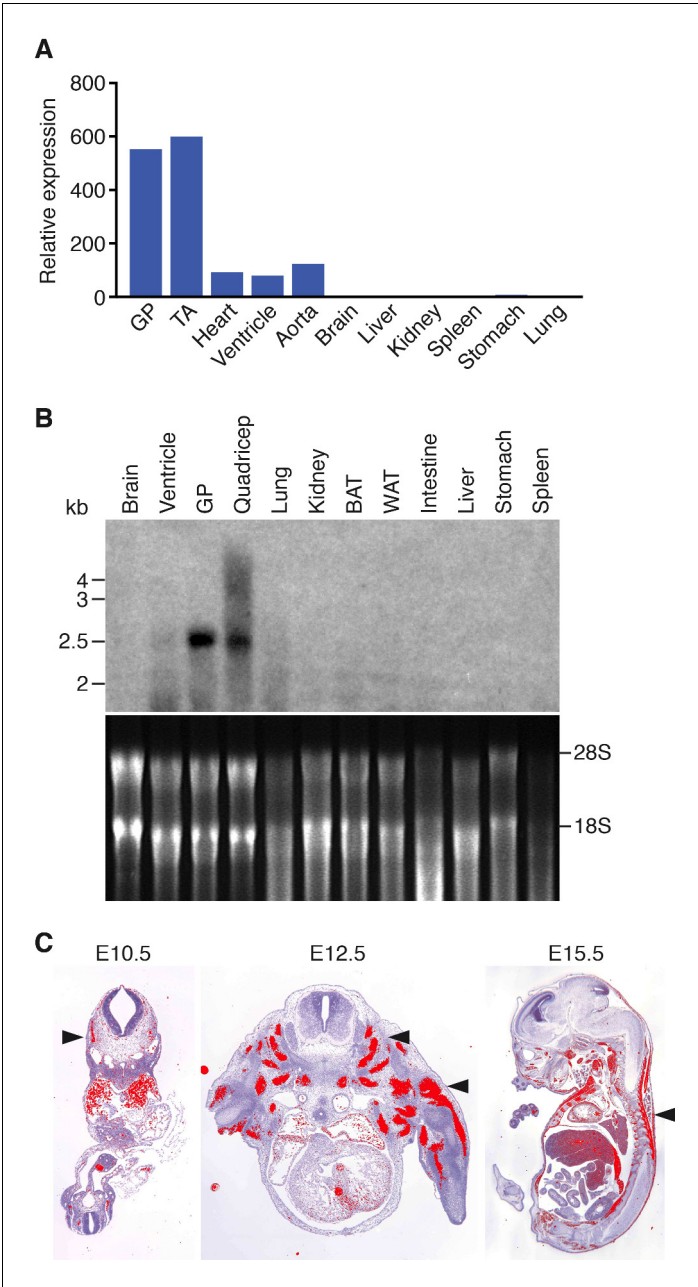

**Figure 1.** Muscle-specific expression of *Klhl41*. (**A**) Distribution of *Klhl41* transcript in adult mouse tissues as determined by qRT-PCR. Expression of *Klhl41* transcript was normalized to 18 s rRNA. GP: Gastrocnemius plantaris, TA: Tibialis anterior. (**B**) Northern blot analysis of *Klhl41* transcript in adult mouse tissues (top). Total RNA in each lane was visualized by ethidium bromide staining and 28 s and 18 s rRNAs were indicated on the right (bottom). BAT: Brown adipose tissue, WAT: White adipose tissue. (**C**) Detection of *Klhl41* transcript by in-situ hybridization using an *Klhl41* anti-sense radioisotopic probe in mouse embryo sections at indicated ages. Transverse sections of E10.5 and E12.5 embryos, and sagittal section of an E15.5 embryo are shown. Signal for *Klhl41 mRNA* (pseudocolored red) only appears in developing muscle. Arrowheads point to representative developing skeletal muscle. The low signal outside the developing muscles represents background.
DOI: https://doi.org/10.7554/eLife.26439.003

The following figure supplement is available for figure 1:

**Figure supplement 1.** *Klhl41* KO allele.
DOI: https://doi.org/10.7554/eLife.26439.004

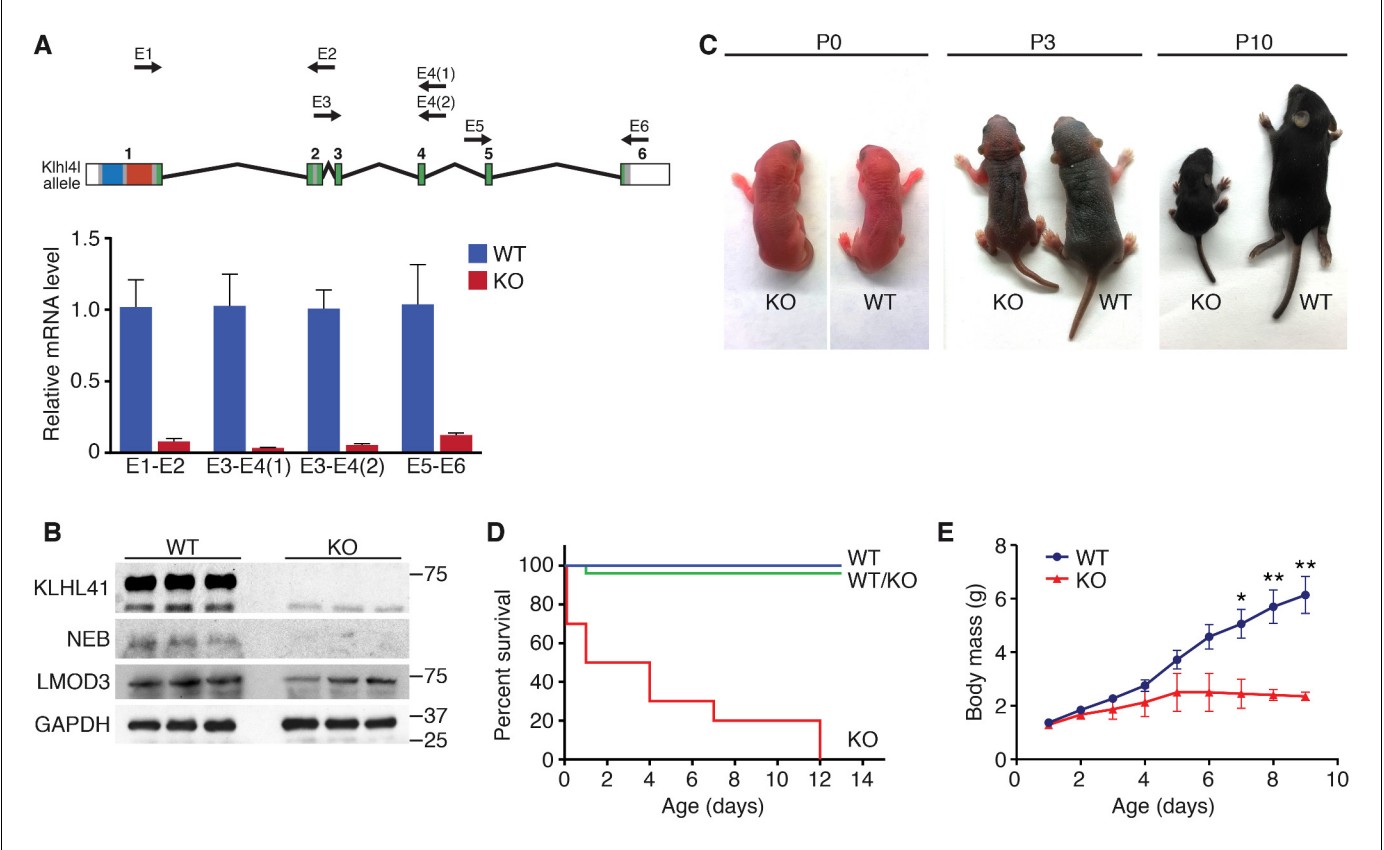

**Figure 2.** *Klhl41* KO mice display neonatal lethality. (**A**) *Klhl41* gene structure: the regions coding for BTB (blue), BACK (red) and Kelch repeats (green) are indicated (top). Detection of *Klhl41* transcript in P0 muscle of WT and *Klhl41* KO mice by qRT-PCR using the indicated primer pairs (arrows) (bottom) (n = 3 mice per genotype). Data are presented as mean ±SEM. (**B**) Western blot analysis of the indicated proteins in WT and KO hindlimb muscles from P0 pups. Because of the large size of NEB (800 > KDa), the corresponding band is above the molecular weight markers used. GAPDH was used as a loading control. (**C**) Surviving WT and *Klhl41* KO mice at P0, P3 and P10. KO pups show a failure-to-thrive phenotype and become severely runted by 10 days of age. (**D**) Survival curve of offspring from *Klhl41* heterozygous intercrosses. n = 20 WT, n = 10 WT/KO (heterozygous), and n = 10 KO. (**E**) Growth curves (body mass) of WT and *Klhl41* KO mice. n = 20 WT and n = 10 KO. Data are presented as mean ±SEM. *p<0.05. **p<0.01.
DOI: https://doi.org/10.7554/eLife.26439.005

supplement 1A*). Remarkably, NEB was the second most down-regulated protein in the muscle of the KO mice (*Figure 4A*). Many other down-regulated proteins in the KO mice were essential components of the sarcomere (*Figure 4B*), including slow skeletal muscle troponin T, myosin light chain-3, myozenin-3, and β-tropomyosin (also associated with NM). As revealed by deep sequencing of RNA transcripts (*Figure 4—source data 2*), the mRNAs encoding these proteins were unchanged in KO muscle. Therefore, the changes in accumulation of these proteins likely reflect post-translational mechanisms. Notably, western blot analysis showed down-regulation of NEB and only a slight decrease in LMOD3 in *Klhl41* KO mice (*Figure 2B*), while both NEB and LMOD3 were markedly down-regulated in *Klhl40* KO mice (*Garg et al., 2014*). The mRNA levels of both proteins did not change in KO muscle, as assessed by RNA-seq and qRT-PCR (*Figure 4—source data 2* and *Figure 4—figure supplement 2C*). We therefore reasoned that KLHL41 could mainly stabilize NEB instead of LMOD3, further underscoring the possibility that these two Kelch proteins act, at least partially, through different mechanisms. Nebulin-related anchoring protein (NRAP), another member of the nebulin family (*Pappas et al., 2011*), was up-regulated both at protein and mRNA levels (probably as a compensatory consequence of sarcomere disarray) (*Figure 4A* and *Figure 4—source data 2*), suggesting that the presence of nebulin repeats is not sufficient for KLHL41 to recognize and stabilize its partners. In contrast to LMOD3, LMOD2, another member of the LMOD family

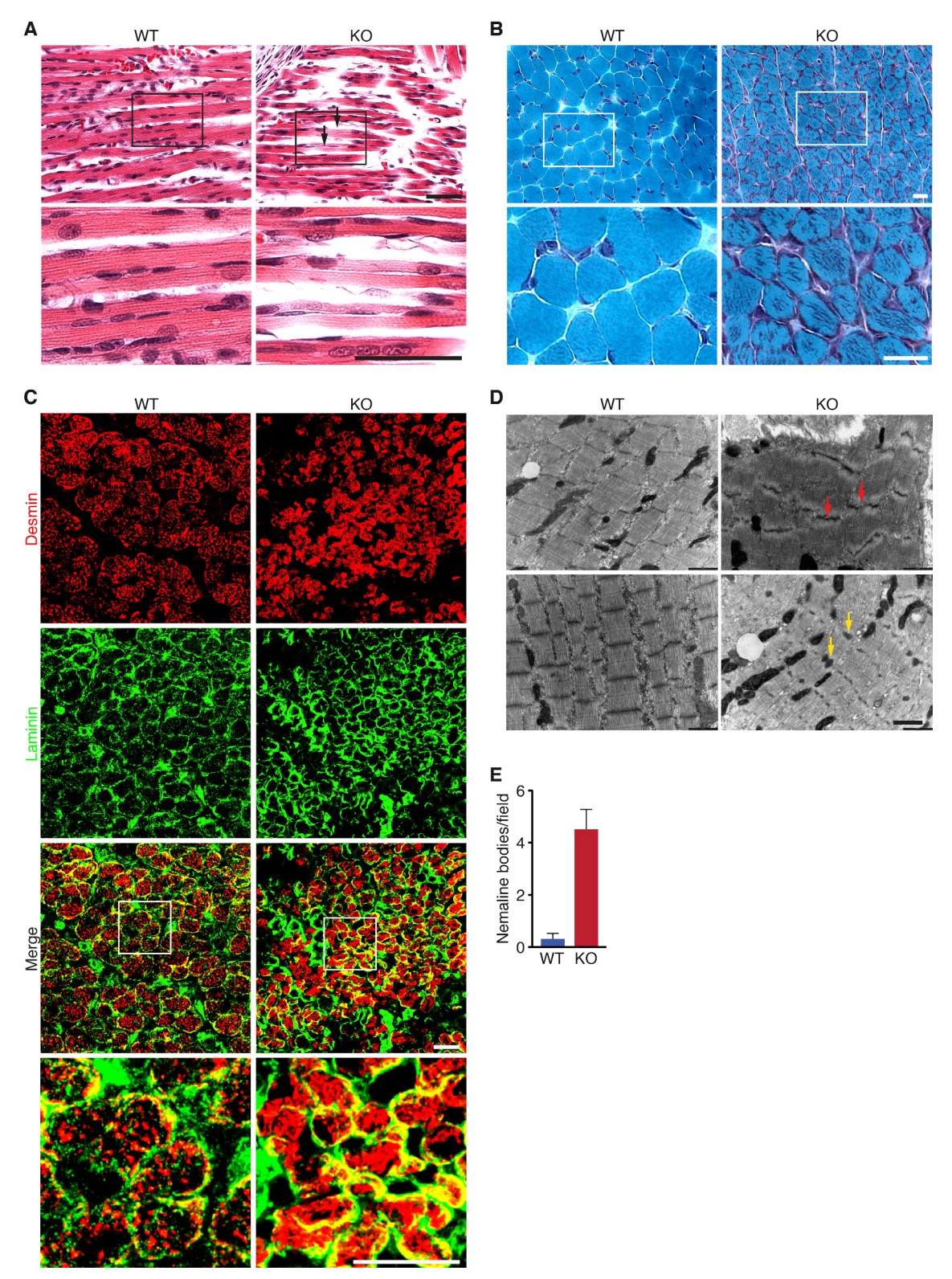

**Figure 3.** Loss of KLHL41 in mice leads to nemaline myopathy. (A) Hematoxylin and eosin staining of longitudinal sections of diaphragm muscle of WT and *Klhl41* KO mice at P0. Arrows point to ragged fibers (fibers with discontinuous H&E staining) present in KO muscle but not WT muscle. Bottom panels are zoomed images of the indicated regions. Scale bar: 20 μm. (B) Gomori's trichrome staining of transverse sections of quadriceps muscle of WT and *Klhl41* KO mice at P10. KO myofibers show numerous abnormal depositions absent in WT muscle. Bottom panels are zoomed images of the
*Figure 3 continued on next page*

*Figure 3 continued*

indicated regions. Scale bar: 20 μm. (**C**) Desmin and laminin immunostaining of transverse sections of WT and *Klhl*41 KO diaphragm at P3. Bottom panels are zoomed images of the indicated regions. Scale bar: 20 μm. (**D**) Electron microscopy images of WT and *Klhl41* KO diaphragm at P3. Z-line streaming and nemaline bodies are indicated with red and yellow arrows, respectively. Scale bar: 1 μm. (**E**) Quantification of nemaline bodies from EM sections of WT and *Klhl41* KO diaphragm at P3. The total number of electrondense protein aggregates (nemaline bodies) was counted for each field of view in WT and KO images (n = 8 fields per genotype). Data are presented as mean ±SEM. p<0.05.
DOI: https://doi.org/10.7554/eLife.26439.006

The following figure supplements are available for figure 3:

**Figure supplement 1.** Additional histology of *Klhl41* KO mice.
DOI: https://doi.org/10.7554/eLife.26439.007

**Figure supplement 2.** Ultrastructural abnormalities in *Klhl41* KO hindlimb.
DOI: https://doi.org/10.7554/eLife.26439.008

involved in thin filament shortening and cardiomyopathy (*Pappas et al., 2015*), was also up-regulated in *Klhl41* KO mice both at protein and mRNA levels.

Among the pathways identified in the up-regulated proteins in KO mice (*Figure 4—figure supplement 1B*), we found 'ubiquitin-dependent catabolic processes'. Indeed, a remarkable number of up-regulated proteins in the KO mice were involved in ubiquitination, including E3 ubiquitin ligases HERC2, TRIM63 and TTC3 (*Figure 4B* and *Figure 4—source data 1*). Among these, TRIM63/MuRF1 has been associated with atrophy and degradation of sarcomeric proteins (*Bodine et al., 2001*). Up-regulation of ubiquitination regulators occurred at both protein and mRNA levels (*Figure 4B* and *Figure 4—source data 2*), suggesting that accumulation of sarcomeric proteins within nemaline bodies might activate compensatory protein degradation pathways. Overall, these results indicate that KLHL41 is required to maintain normal levels of sarcomeric proteins in vivo.

## KLHL41 and KLHL40 show distinct substrate preference despite their high homology

In contrast to other Kelch proteins, KLHL40 stabilizes its two main binding partners (NEB and LMOD3) instead of degrading them (*Garg et al., 2014*). Due to the high similarity between KLHL40 and 41 (*Figure 4—figure supplement 3A*), we tested the stabilization of NEB and LMOD3 by KLHL41 in transfected COS-7 cells (*Figure 4C*). Because of the large size of NEB (>800 KDa), which prohibits efficient expression of the full-length protein, we used a fragment of the NEB protein (NEB$_{frag}$) previously found to associate with KLHL40 in yeast two-hybrid assays (*Garg et al., 2014*). NEB$_{frag}$ alone could not be detected by western blot, but NEB$_{frag}$ protein levels were stabilized when either KLHL40 or KLHL41 were co-expressed (*Figure 4C*). In contrast, LMOD3 levels only increased modestly in the presence of KLHL41, whereas KLHL40 overexpression was sufficient to dramatically increase LMOD3 stability (*Figure 4C*). These results indicate that KLHL41 preferentially stabilizes NEB$_{frag}$ over LMOD3, confirming the functional distinctions between KLHL40 and KLHL41.

## KLHL41 interacts with numerous sarcomeric proteins involved in myopathy

To explore the molecular functions of KLHL41, we performed tandem affinity purification (TAP) of 3xFLAG-HA tagged KHL41 in C2C12 myotubes and identified KHL41 binding partners by mass spectrometry (*Figure 4D*). Structural components of the sarcomere, such as NEB, NRAP and filamin-C (FLNC) were identified. Notably, mutations in *NEB*, *NRAP*, and *FLNC* have been associated with NM and other myopathies (*D'Avila et al., 2016*; *Duff et al., 2011*). KLHL40 was also identified as a KLHL41 binding partner, suggesting a heterodimeric function of these proteins.

We further validated the interaction between KLHL41 and both NEB$_{frag}$ and FLNC by co-immunoprecipitation in COS-7 cells (*Figure 4E* and *Figure 4—figure supplement 3B*). We did not observe changes in either NRAP or FLNC protein levels when KLHL41 was co-expressed, indicating that the stabilizing activity of KLHL41 may be partner specific. Overall, these results suggest that KLHL41 interacts with components of the sarcomere and loss of these interactions leads to sarcomeric disarray and NM. LMOD3 was not detected by TAP using 3xFLAG-HA-KLHL41, nor did we detect an interaction between these proteins in co-immunoprecipitation experiments (*Figure 4—figure*

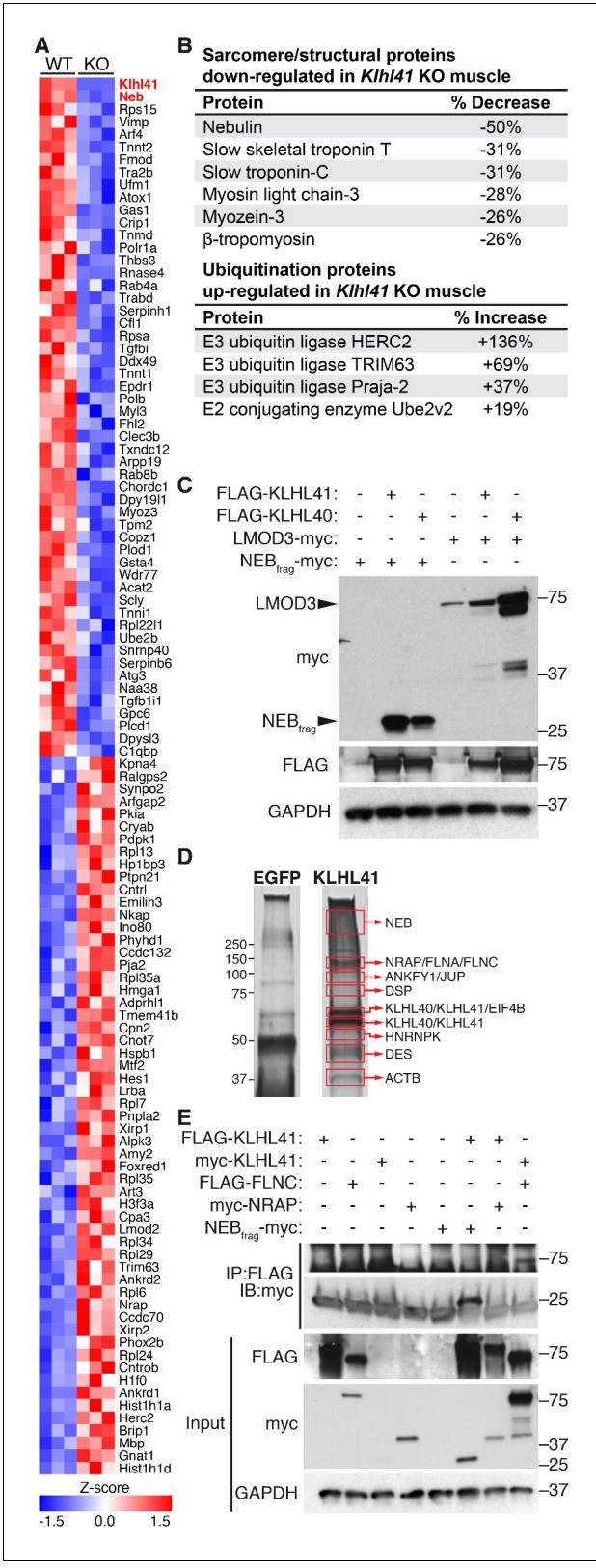

**Figure 4.** KLHL41 is required to maintain normal levels of sarcomeric proteins in vivo. (**A**) Heat map of changes in protein levels identified by proteomics in WT and *Klhl41* KO hindlimb muscles at P0 (n = 3 mice per genotype). A total of 389 proteins were identified by proteomics to be differentially regulated (p<0.05). Proteins up or down-regulated by more than 30% compared to WT are represented (n = 111). KLHL41 and NEB (red) were the two

*Figure 4 continued on next page*

*Figure 4 continued*

most down-regulated proteins. Heat map was created with Morpheus and the color scale represents Z-score. (**B**) Selected list of down-regulated sarcomeric proteins and up-regulated ubiquitination proteins identified by proteomics. Their relative levels to WT are shown on the right. (**C**) Effect of KLHL41 on NEB$_{frag}$ and LMOD3 stability. KLHL41 preferentially stabilizes NEB$_{frag}$. COS-7 cells were transfected with expression vectors for the indicated proteins. Protein levels of LMOD3, NEB$_{frag}$, KLHL40 and KLHL41 were detected by western blot. GAPDH was used as a loading control. (**D**) KLHL41 binding partners identified following tandem affinity purification (TAP) from cultured C2C12 myotubes. Representative silver-stained gels with TAP protein from myotubes infected with 3XFLAG-HA-EGFP (negative control) or 3xFLAG-HA-KLHL41 are shown. Proteins listed next to each box indicate the most abundant protein(s) identified in each corresponding area. (**E**) Co-immunoprecipitation experiments to validate interactions between KLHL41 and FLNC, NRAP and NEB$_{frag}$. COS-7 cells were transfected with the indicated plasmids.

DOI: https://doi.org/10.7554/eLife.26439.009

The following source data and figure supplements are available for figure 4:

**Source data 1.** Global protein changes in WT and KO mice by quantitative proteomics.
DOI: https://doi.org/10.7554/eLife.26439.013
**Source data 2.** Global mRNA changes in WT and KO mice by RNA-seq.
DOI: https://doi.org/10.7554/eLife.26439.014
**Figure supplement 1.** Pathway analysis of differentially regulated proteins in *Klhl41* KO muscle by unbiased proteomics.
DOI: https://doi.org/10.7554/eLife.26439.010
**Figure supplement 2.** Global changes in mRNA expression by RNA-seq.
DOI: https://doi.org/10.7554/eLife.26439.011
**Figure supplement 3.** KLHL40 and KLHL41 show distinct partner specificity.
DOI: https://doi.org/10.7554/eLife.26439.012

*supplement 3B*). These results indicate that despite their similarity in structure, KLHL41 and KLHL40 have both common and distinct partners that may contribute to NM through different mechanisms.

## The BTB domain of KLHL41 interacts with components of the CUL3 complex

Kelch proteins are known to associate with each other and to form complexes with CUL3 (*Dhanoa et al., 2013*). We reasoned that the overlapping partners between KLHL41 and KLHL40 could reflect their association with a common complex. Indeed, in co-immunoprecipitation assays, we found that KLHL41 self-associated and interacted with KLHL40 (*Figure 5A*), as well as with CUL3 (*Figure 5B*). To define the domains required for these interactions, we created deletion mutants lacking each of the three annotated domains of KLHL41 (BTB, BACK and Kelch repeats). By co-immunoprecipitation, we observed that deletion of the BTB domain was sufficient to abolish homo-dimerization of KLHL41 (*Figure 5C*) and its association with CUL3 (*Figure 5D*). Similarly, deletion of the BTB domain of KLHL40 greatly impaired its association with KLHL41 (*Figure 5E*). Overall, these experiments indicate that the BTB domain of KLHL41 constitutes a critical region for interaction with other components of the CUL3 complex.

## Poly-ubiquitination is required for KLHL41 stabilizing activity

Most Kelch proteins act as adaptors that confer substrate specificity to E3 ubiquitin ligase complexes (*Gupta and Beggs, 2014*). To assess whether poly-ubiquitination could regulate NEB$_{frag}$ accumulation, we tested the effect of HA-tagged ubiquitin mutants on NEB$_{frag}$ protein levels. The formation of poly-ubiquitination chains typically occurs via the covalent attachment of the 76-amino acid ubiquitin peptide to lysine residues of proteins that are targeted for degradation (*Komander and Rape, 2012*). The HA-tagged Ubiquitin-K0 (lysine zero) (Ub-K0) is a mutant protein that contains all its lysines mutated to arginines, thus preventing poly-ubiquitination in a dominant negative manner (*Kulathu and Komander, 2012*). Surprisingly, overexpression of Ub-K0 was sufficient to prevent the accumulation of NEB$_{frag}$ in the presence of KLHL41 (*Figure 6A*). In contrast, overexpression of Ub-K0 did not affect LMOD3 stabilization by KLHL40 (*Figure 6—figure supplement 1*). These findings

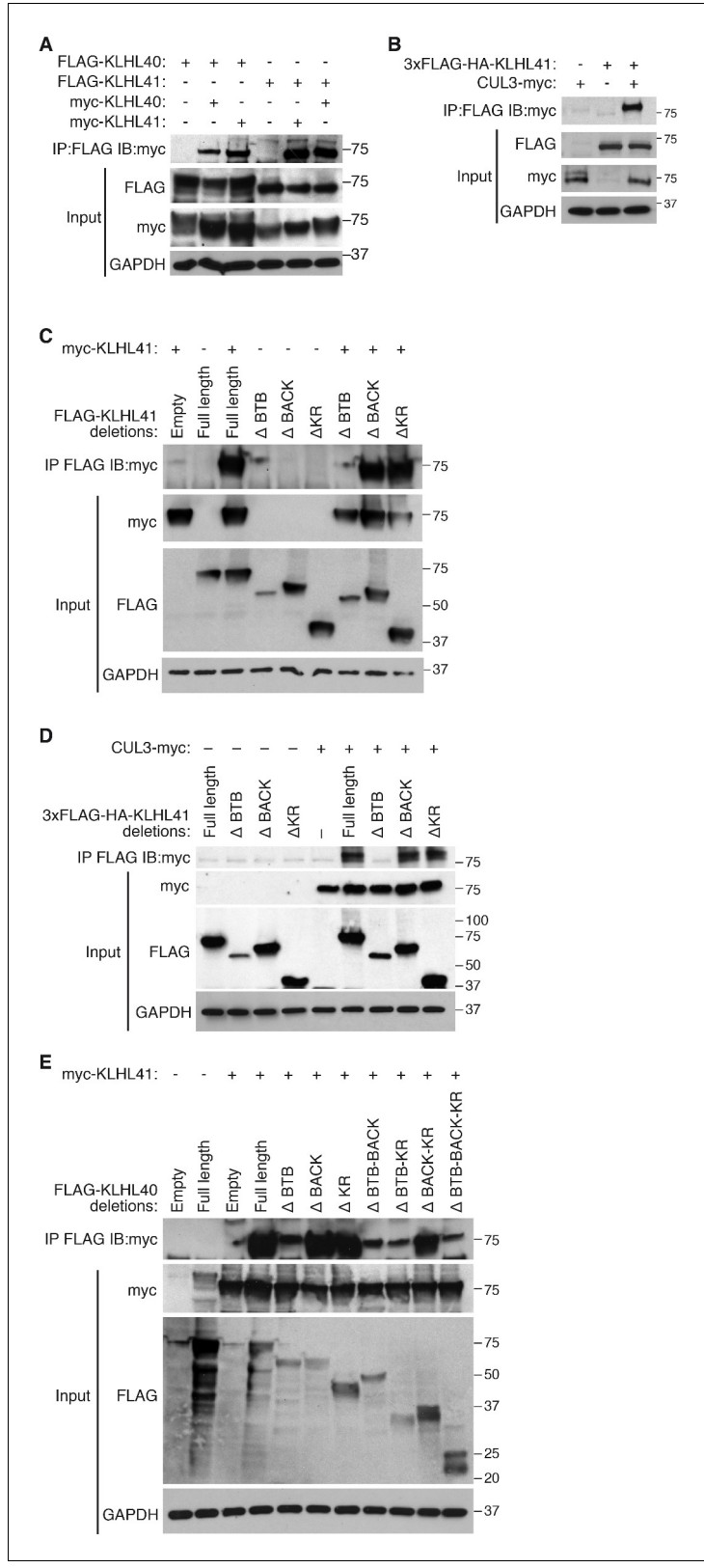

**Figure 5.** KLHL41 association with CUL3 is mediated by its BTB domain. (**A**) Co-immunoprecipitation in transfected COS-7 cells indicates that KLHL41 homodimerizes with itself and heterodimerizes with KLHL40. Western blots of co-immunoprecipitation and input with indicated antibodies are shown. GAPDH was used as a loading control. (**B**) Interaction between 3XFLAG-HA-KLHL41 and CUL3-myc was detected by co-

*Figure 5 continued on next page*

*Figure 5 continued*

immunoprecipitation and western blot analysis with the indicated antibodies in transfected COS-7 cells. (**C**) Co-immunoprecipitation of various FLAG-tagged KLHL41 deletion mutants with full-length KLHL41 (myc-KLHL41) in transfected COS-7 cells to identify the domains necessary for KLHL41 self-dimerization. Deletion of the BTB domain (ΔBTB) abolished the interaction with full-length KLHL41. (**D**) Co-immunoprecipitation of various FLAG-tagged KLHL41 deletion mutants with CUL3-myc in transfected COS-7 cells to identify the domains necessary for association with CUL3. Deletion of the BTB domain (ΔBTB) abolished the interaction with CUL3-myc. (**E**) Co-immunoprecipitation of various FLAG-tagged KLHL40 deletion mutants with full-length KLHL41 (myc-KLHL41) in transfected COS-7 cells to identify the domains of KLHL40 required for heterodimerization with KLHL41. All FLAG-KLHL40 mutants in which the BTB domain of KLHL40 was deleted (ΔBTB; ΔBTB-BACK; ΔBTB−KR and ΔBTB−BACK−KR) presented diminished interaction with the full-length myc-KLHL41.
DOI: https://doi.org/10.7554/eLife.26439.015

suggest that poly-ubiquitination was unexpectedly required for the stabilization of $NEB_{frag}$ by KLHL41.

Next, we sought to identify the protein target of poly-ubiquitination. Overexpression of Ub-K0 collapsed high molecular bands of KLHL41 corresponding to a potentially poly-ubiquitinated pool (*Figure 6B*), suggesting that KLHL41 was the target of poly-ubiquitination. Co-immunoprecipitation experiments showed that KLHL41 was poly-ubiquitinated and overexpression of Ub-K0 greatly reduced KLHL41 poly-ubiquitination (*Figure 6C*). Poly-ubiquitination can take place through any of the 7 lysine residues of ubiquitin or the free amino group from the first methionine. To understand which type of ubiquitination regulated KLHL41, we used HA-tagged ubiquitin mutants in which individual lysines were mutated to arginine (K6R, K11R, K27R, K29R, K33R, K48R and K63R). We found that overexpression of K48R reduced $NEB_{frag}$ stabilization by KLHL41, whereas other mutants did not affect $NEB_{frag}$ levels (*Figure 6D*).

To identify the region of KLHL41 that is sensitive to poly-ubiquitination, we co-expressed KLHL41 deletion mutants with Ub-WT or Ub-K0 and observed that deletion of the BTB domain was sufficient to strongly impair KLHL41 poly-ubiquitination (*Figure 6C*). The loss of $NEB_{frag}$ stability in the presence of Ub-K0 (*Figure 6A*) suggested that reduced poly-ubiquitination could inhibit KLHL41 activity. Based on the preferential poly-ubiquitination of the BTB domain, we tested whether deletion of any functional domain of KLHL41 could rescue $NEB_{frag}$ levels even in the presence of Ub-K0. As previously reported for KLHL40, deletion of the Kelch repeats of KLHL41 only decreased $NEB_{frag}$ protein levels slightly when Ub-WT was overexpressed (*Garg et al., 2014*). However, when poly-ubiquitination was inhibited by Ub-K0, ΔBTB-KLHL41 could still stabilize $NEB_{frag}$ (*Figure 6E*). While components of the E3 ligase complex are usually ubiquitinated (*Fang et al., 2000*; *Nuber et al., 1998*), the requirement of the BTB domain for poly-ubiquitin sensitivity suggests that ubiquitination plays a direct role in the regulation of KLHL41 activity. Therefore, we conclude that poly-ubiquitination of the BTB domain of KLHL41 mediates $NEB_{frag}$ stabilization.

## KLHL41 prevents aggregation of $NEB_{frag}$

Because mutations in NEB are the most common cause of NM (*Romero et al., 2013*), we investigated the mechanism by which KLHL41 stabilizes NEB. Cycloheximide chase experiments showed that $NEB_{frag}$ had a short half-life of ~6 hr (*Figure 7—figure supplement 1A*). We hypothesized that KLHL41 could either prevent degradation of NEB by another E3 ligase or act as a chaperone. Inhibition of proteasomal activity or autophagy by MG132 or chloroquine, respectively, was not sufficient to rescue $NEB_{frag}$ protein levels in the absence of KLHL41 (*Figure 7A–B*), suggesting that KLHL41 did not evoke its stabilizing effect on $NEB_{frag}$ by degrading another E3 ligase.

Next, we sought to identify a potential degron in $NEB_{frag}$ that could target it for degradation. We therefore generated a series of $NEB_{frag}$ mutants each containing a 25-residue deletion spanning the entire protein (*Figure 7—figure supplement 1B–C*), and assessed their protein levels by western blot in the presence or absence of KLHL41. Strikingly, we observed that in the absence of KLHL41, and in contrast to the full length $NEB_{frag}$, most deletion mutants could be detected even in the absence of KLHL41, albeit at lower levels than with co-expression of KLHL41 and the full length $NEB_{frag}$ (*Figure 7C*, up). Furthermore, in the presence of KLHL41, all $NEB_{frag}$ deletion mutants exhibited increased protein levels compared to the full length $NEB_{frag}$ and they interacted with KLHL41 by co-

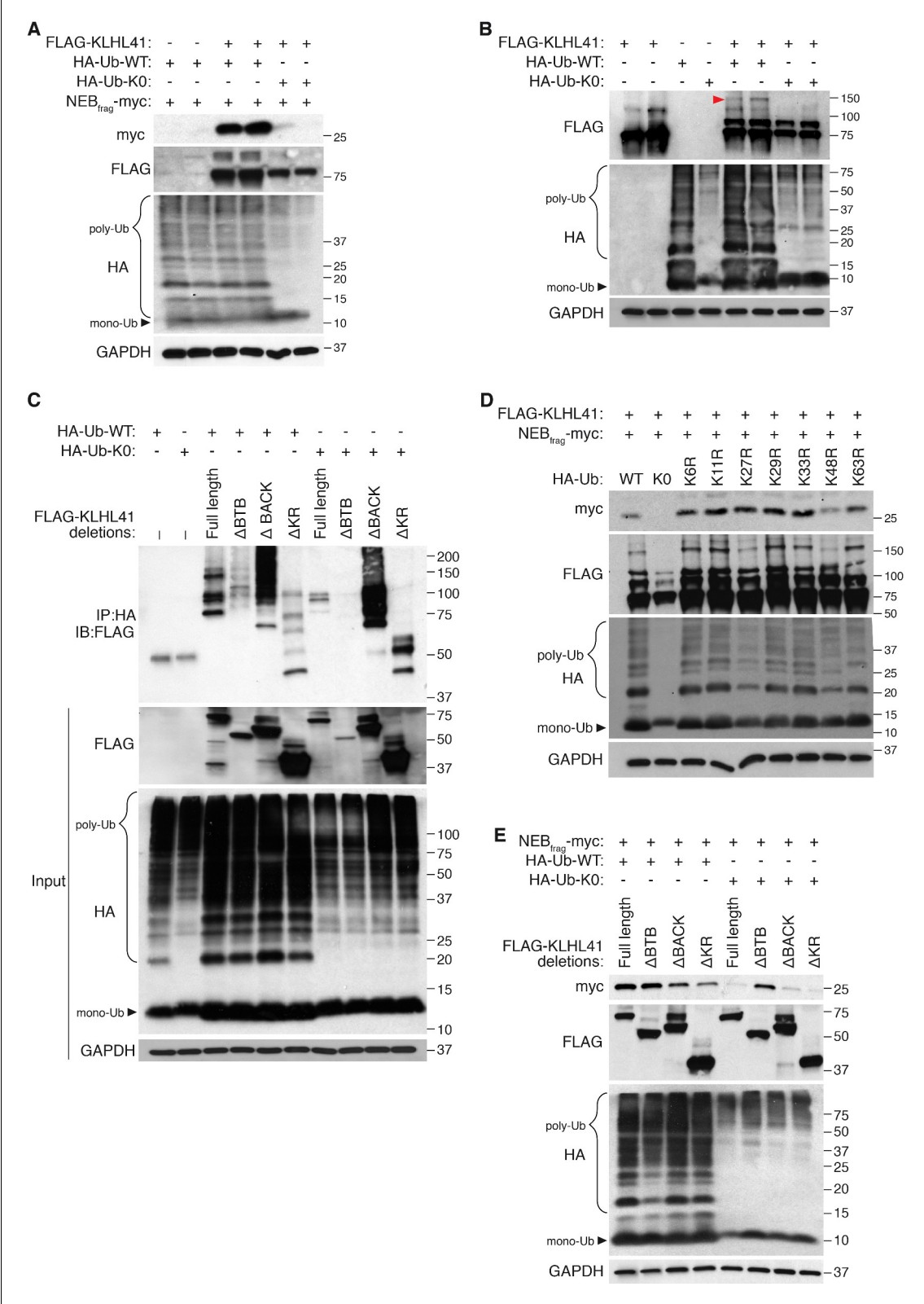

**Figure 6.** KLHL41 stabilization of NEB_frag is poly-ubiquitin-dependent. (**A**) KLHL41 can stabilize NEB_frag in the presence of HA-Ub-WT but not HA-Ub-K0 mutant. Expression of NEB_frag and KLHL41 was determined by western blot analysis with the indicated antibodies in COS-7 cells. The laddered smear detected in HA input with HA-Ub-WT corresponds to the pool of poly-ubiquitinated proteins (indicated as poly-Ub). In contrast, HA-Ub-K0 (lysine zero), a mutant protein that cannot be poly-ubiquitinated, was preferentially detected as a single band of mono-Ubiquitin (indicated as mono-Ub). Note that

*Figure 6 continued on next page*

*Figure 6 continued*

HA-Ub-K0 prevented stabilization of NEB$_{frag}$ by KLHL41. (B) Western blot analysis of FLAG-KLHL41 showed high molecular bands when expressed alone in COS-7 cells. Those bands were further increased when HA-Ub-WT was co-expressed (red arrowhead). Overexpression of HA-Ub-K0 collapsed the high molecular bands of FLAG-KLHL41, indicating that they correspond to a pool of poly-ubiquitinated KLHL41. (C) KLHL41 is preferentially poly-ubiquitinated in the BTB domain. Full-length and deletion mutants of KLHL41 were co-expressed in the presence of HA-Ub-WT or HA-Ub-K0 in COS-7 cells. Protein extraction was performed in the presence of deubiquitinase inhibitors and protein expression was detected by western blot analysis using the indicated antibodies. The laddered pattern in the IP:HA IB:FLAG western blot corresponds to poly-ubiquitinated KLHL41. Note that either deletion of BTB (ΔBTB) or HA-Ub-K0 overexpression collapsed poly-ubiquitinated bands. (D) HA-Ub-K48R impairs KLHL41 activity to stabilize NEB$_{frag}$. KLHL41 stabilization of NEB$_{frag}$ in the presence of Ubiquitin HA-Ub-WT, K0 and Ub lysine mutants were determined by western blot analysis. HA-Ub-K48R overexpression led to a decrease in poly-ubiquitinated KLHL41, as observed in the FLAG input western blot. Each Ub lysine mutant only inhibits one type of ubiquitination. Therefore, total poly-ubiquitination levels remain the same as HA-Ub-WT as detected in HA input western blot. (E) Deletion of the KLHL41 BTB domain (ΔBTB) can restore NEB$_{frag}$ stability in the presence of Ub-K0 mutant. Western blot analysis shows NEB$_{frag}$ stability in the presence of full length and deletion mutants of KLHL41 when co-expressed with either HA-Ub-WT or HA-Ub-K0 mutant in COS-7 cells.

DOI: https://doi.org/10.7554/eLife.26439.016

The following figure supplement is available for figure 6:

**Figure supplement 1.** KLHL40 stabilization of LMOD3 is independent of poly-ubiquitination.

DOI: https://doi.org/10.7554/eLife.26439.017

immunoprecipitation (d5 and d6 had weaker interactions than others) (*Figure 7C*, bottom). NEB$_{frag}$ contains multiple nebulin repeats enriched in basic and aromatic amino acids (*Figure 7—figure supplement 1B–C*). There are no apparent sequence differences among deletion mutants that could explain the different activities of d5 and d6 from other mutants (*Figure 7—figure supplement 1C*). The fact that multiple deletion mutants could interact with KLHL41 suggests that the nebulin repeats present in NEB$_{frag}$ may be sufficient for the interaction. These findings indicate that NEB$_{frag}$ is unlikely to be regulated by a degron recognized by an unknown E3 ligase but rather general conformational changes promote NEB$_{frag}$ stability.

To assess whether KLHL41 prevented NEB$_{frag}$ aggregation, we performed protein extraction of the insoluble fraction of transfected COS-7 cells with high detergent concentration. Surprisingly, we found that when NEB$_{frag}$ was expressed alone, it could be detected in the insoluble fraction, whereas co-expression of KLHL41 was sufficient to shift a portion of the NEB$_{frag}$ pool to the soluble fraction (*Figure 7D*). We further validated these results by immunofluorescence (*Figure 7E* and *Figure 7—figure supplement 1D*). NEB$_{frag}$ alone formed aggregates localized predominantly in the nucleus, and co-expression of KLHL41 or KLHL40 resulted in homogenous cytosolic staining. Thus, these results suggest that KLHL41 and KLHL40 can act as chaperones and prevent NEB$_{frag}$ aggregation.

## Discussion

Mutations in KLHL41 have been associated with NM in humans, but molecular understanding of the mechanism of action of KLHL41 has been lacking, despite its clinical significance. Furthermore, the unique stabilizing activity of both KLHL41 and KLHL40, which distinguishes them from most other members of the Kelch family, has not been previously explored. Our results demonstrate that KLHL41 is required for sarcomere integrity and stabilization of essential muscle structural proteins. The absence of KLHL41 in mice results in severe NM with general sarcomere disarray, accumulation of nemaline bodies and perinatal death as seen in humans with KLHL41 mutations. Our results reveal a unique pro-stabilizing function of poly-ubiquitinated KLHL41 in which it prevents NEB aggregation. A model consistent with our findings is shown in *Figure 7F*. We surmise that under normal conditions KLHL41 functions as a chaperone, preventing NEB aggregation and degradation. Loss of KLHL41 or reduced poly-ubiquitination of KLHL41 results in loss of KLHL41 activity, NEB aggregation and NM. These regulatory events suggest that KLHL41 may serve as a poly-ubiquitination sensor, offering a new layer of complexity to the abundant ubiquitin ligases and deubiquitinases resident in muscle sarcomeres.

### KLHL41 is a poly-ubiquitin dependent chaperone

Protein ubiquitination involves the covalent attachment of the 76-amino acid ubiquitin peptide to the epsilon amino group of target lysine residues. Poly-ubiquitin chains can be formed by sequential

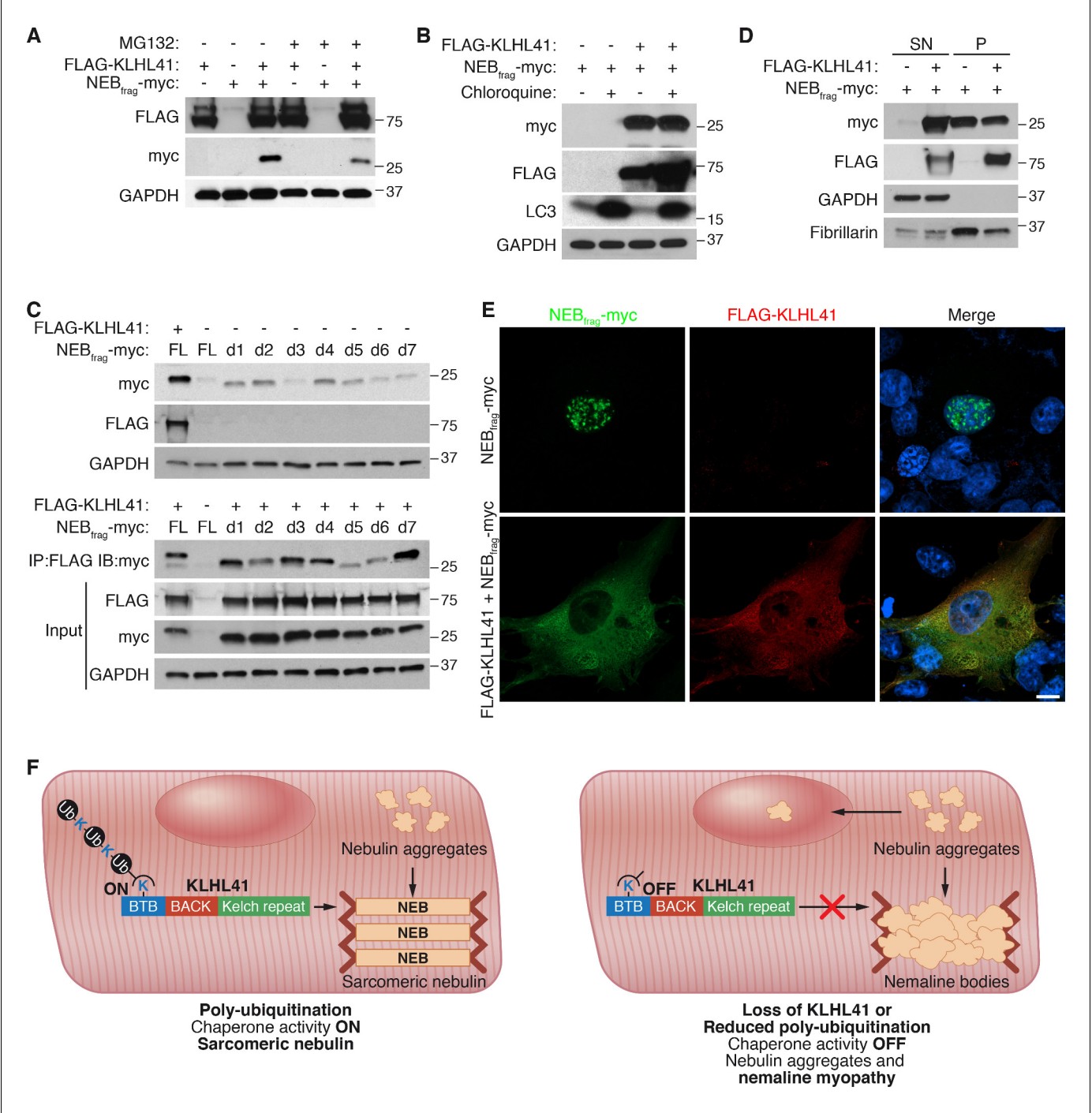

**Figure 7.** KLHL41 prevents NEB<sub>frag</sub> aggregation in the nucleus. (**A**) KLHL41 regulation of NEB$_{frag}$ in the absence or presence of proteasome inhibitor (10 μM MG132 for 24 hr) was determined by western blot analysis with the indicated antibodies in COS-7 cells. Note that DMSO treated cells were used as negative (-) control. Expression of NEB$_{frag}$ was not detected in the presence of MG132, and KLHL41 stabilizes NEB$_{frag}$ in the presence or absence of MG132. (**B**) KLHL41 regulation of NEB$_{frag}$ in the absence or presence of autophagy inhibitor (10 μM chloroquine for 12 hr) in COS-7 cells. Note that DMSO treated cells were used as negative (-) control. Expression of NEB$_{frag}$ was not detected in the presence of chloroquine, and KLHL41 stabilizes NEB$_{frag}$ in the presence or absence of chloroquine. Increased level of LC3 protein was detected as a positive control for autophagy inhibition. (**C**) Multiple NEB$_{frag}$ deletion mutants are stable in the absence of KLHL41. COS-7 cells were transfected with various NEB$_{frag}$ deletion mutants in the absence (top) or presence (bottom) of KLHL41. Protein levels were determined by western blot analysis with the indicated antibodies. Co-immunoprecipitation of FLAG-KLHL41 and NEB$_{frag}$ deletion mutants (bottom) shows that all mutants interact with KLHL41. (**D**) Expression of KLHL41 affects NEB$_{frag}$ solubility. COS-7 cells were transfected with the indicated plasmids and lysed with mild detergent to collect supernatant (SN). The
*Figure 7 continued on next page*

*Figure 7 continued*

remaining pellet (P) was then solubilized under high detergent conditions. Expression of FLAG-KLHL41 and NEB$_{frag}$ in SN and P fractions was detected by western blot analysis. In the absence of KLHL41, NEB$_{frag}$ was exclusively expressed in P. However, in the presence of KLHL41, total NEB$_{frag}$ levels were increased and enriched in the SN fraction. GAPDH was restricted to the SN fraction. Fibrillarin, a nucleolar protein, was enriched in P. (E) Expression of NEB$_{frag}$ in the absence and presence of KLHL41 was detected by immunofluorescence of COS-7 cells transfected with the indicated plasmids. Note that NEB$_{frag}$ was accumulated in the nucleus in the absence of KLHL41 but it was redistributed to the cytoplasm when KLHL41 was co-expressed. Scale bar: 10 µm. (F) Model for regulation of KLHL41 stabilizing activity. KLHL41 acts as a poly-ubiquitin dependent chaperone that prevents NEB aggregation. Loss of KLHL41 or inhibition of its poly-ubiquitination leads to NEB aggregation, sarcomere disarray and nemaline myopathy.
DOI: https://doi.org/10.7554/eLife.26439.018

The following figure supplement is available for figure 7:

**Figure supplement 1.** KLHL41 prevents NEB$_{frag}$ aggregation in the nucleus.
DOI: https://doi.org/10.7554/eLife.26439.019

addition of ubiquitin molecules through any of its 7 lysines (K6, K11, K27, K29, K33, K48 and K63) or the free amino group of the initial methionine (*Yau and Rape, 2016*). K48-linked poly-ubiquitination has been extensively characterized as the canonical signal for proteasomal degradation and most Kelch proteins whose functions have been explored mediate protein degradation via E3 ligase-dependent poly-ubiquitination (*Furukawa and Xiong, 2005*; *Liu et al., 2016*; *Shibata et al., 2013*; *Xu et al., 2003*). However, new nonproteolytic roles for Kelch proteins are also being reported. For example, KLHL20 controls trafficking of the actin stabilizing protein Coronin 7 via K33-linked poly-ubiquitination (*Yuan et al., 2014*), while mono-ubiquitination NOLC1 and TCOLF by KBTBD8 regulates translation and cell fate specification (*Werner et al., 2015*).

Our findings highlight a unique protein stabilizing function of KLHL41. In contrast to other Kelch proteins, KLHL41 stabilizes its partner, NEB, instead of marking it for degradation. Inhibition of the two major degradation pathways (the ubiquitin-proteasome and the lysosome) was not sufficient to rescue NEB$_{frag}$ levels in the absence of KLHL41, suggesting that KLHL41 does not regulate another E3 ligase responsible for NEB degradation. Intriguingly, inhibition of poly-ubiquitination abolished the ability of KLHL41 to stabilize NEB$_{frag}$. Additionally, we could not identify a degron responsible for NEB$_{frag}$ instability in the absence of KLHL41 but rather, multiple mutants were stable when short sequences of the protein were deleted. These results suggest that KLHL41 stabilizes NEB$_{frag}$ by regulating its folding rather than by degrading an unknown protein that might decrease NEB$_{frag}$ levels. Specific inhibition of K48-linked poly-ubiquitination decreased NEB$_{frag}$ stabilization by KLHL41. Besides its canonical role in degradation, K48-linked poly-ubiquitination has been implicated in substrate stabilization (*Flick et al., 2006*) and segregation of ubiquitinated proteins from their partners (*Ramadan et al., 2007*; *Rape et al., 2001*). In contrast to those studies, however, KLHL41 itself is ubiquitinated instead of its partner. Components of the E3 ubiquitin ligase complex can be self-ubiquitinated during the transfer of ubiquitin to their substrates or as a negative feedback loop to down-regulate their protein levels (*Fang et al., 2000*; *Nuber et al., 1998*), but the requirement of Kelch protein poly-ubiquitination for protein stabilization constitutes a novel regulatory mechanism.

We speculate that K48-linked poly-ubiquitination of KLHL41 could regulate protein-protein interactions between the BTB domain and other partners. K48-linked poly-ubiquitination can act as a recognition signal for CDC48, a chaperone required to extract misfolded proteins from the ER during ER-associated protein degradation (*Jentsch and Rumpf, 2007*). Additional work will be required to understand how KLHL41 activity is regulated in a poly-ubiquitin dependent manner.

## KLHL40 and KLHL41 stabilize their partners via distinct mechanisms

Although KLHL40 and 41 share extensive homology, and loss of either gene causes NM, our results indicate that these two proteins possess both overlapping and distinct functions. Loss of either KLHL40 or KLHL41 leads to severe NM in humans, whereas *Klhl41* KO mice present earlier onset of muscle dysfunction than *Klhl40* KO mice. While both proteins stabilize NEB, KLHL41 stabilizes LMOD3 at lower levels than KLHL40. Considering that the loss of function phenotype of KLHL41 is stronger than that of KLHL40, it is likely KLHL41 is more critical for sarcomere integrity by interacting with additional partners.

The stabilization of LMOD3 by KLHL40 occurs through a mechanism distinct from the stabilization of NEB, as inhibition of poly-ubiquitination does not decrease LMOD3 protein levels in the presence

of KLHL40 (*Garg et al., 2014*). Furthermore, in the absence of KLHL40, LMOD3 levels are increased by proteasome inhibition. KLHL40 and KLHL41 are highly similar in their BTB and BACK domains, which could explain why both are sensitive to poly-ubiquitination. However, they present differences within the Kelch repeats, which likely enables them to discriminate between different substrates. It is currently unknown whether other Kelch proteins, especially those closely related to KLHL41, act as chaperones or can stabilize their partners in a poly-ubiquitin dependent manner. Of special interest is KBTBD13, the only other Kelch protein associated with NM. Further work is needed to understand if KBTBD13 acts through a similar mechanism to that of KLHL40 and KLHL41.

## Therapeutic implications

NM and other protein aggregation myopathies are characterized by formation of pathogenic protein inclusions (*Goebel and Blaschek, 2011*). Although the most common causes for protein aggregation in myofibers are mutations in sarcomeric genes, mutations in the *MuRF1* and *MuRF3* E3 ubiquitin ligases have also been reported (*Olivé et al., 2015*). Nemaline bodies can be detected in the cytoplasm and the nucleus of patient biopsies, and the presence of intranuclear rods has been associated with more severe clinical phenotypes (*Ryan et al., 2003*). Chaperones have become interesting therapeutic targets in protein aggregation diseases (*Smith et al., 2014*; *Winter et al., 2014*). Indeed, during myofibrillogenesis, chaperones are required for proper assembly of the sarcomere and loss of their activity has been associated with a broad array of muscle disorders (*Sarparanta et al., 2012*; *Selcen et al., 2009*). Our results reveal an unexpected chaperone activity for KLHL40 and KLHL41, suggesting that modulation of chaperone activity could represent an approach to treat NM. There are currently no effective therapies for NM. Thus, elucidation of the precise mechanisms of action of KLHL40 and KLHL41 may ultimately allow new interventions into the pathogenic processes associated with this disorder.

## Materials and methods

### Generation of *Klhl41* KO mice

All experimental procedures involving animals in this study were reviewed and approved by the University of Texas Southwestern Medical Center's Institutional Animal Care and Use Committee. Mice were generated using a targeted *Klhl41* embryonic stem cell clone (*Klhl41*^tm1a(KOMP)Wtsi^) obtained from the KOMP Repository (http://www.KOMP.org) as previously described (*Millay et al., 2013*). *Klhl41*^+/−^ mice were intercrossed to generate KO mice. KO mice were maintained in a pure C57BL/6 background. *Klhl40* KO mice have been previously described (*Garg et al., 2014*).

### Genotyping of *Klhl41 and Klhl40 KO* mice

*Klhl41* and *Klhl40* genotypes were determined based on the presence or absence of WT and KO alleles using two genotyping reactions. The primers for *Klhl41* span intron 1 (WT) or the targeted allele plus intron 1 (KO). The primers for *Klhl40* span exon 1 and a region removed in the targeted allele (WT) or the LacZ cassette and a region outside of the KO allele (KO). The following primers were used:

| Primer | Sequence (5' to 3') |
| --- | --- |
| KLHL41-WT-F | AGAAAGTAAGTGCCAAAATGAATCC |
| KLHL41-WT-R | AGGCTGACTGTGCTCCTAGGTGCTGTTC |
| KLHL41-KO-F | GAGATGGCGCAACGCAATTAAT |
| KLHL41-KO-R | CAGTTTCTCGTTCAGTTCTTCTCTG |
| KLHL40-WT-F | TATATATAGCCCAGGGCAGACAG |
| KLHL40-WT-R | CGCACTACACAGTCTAGGAACTTG |
| KLHL40-KO-F | TCGAGAGACCTTCCAGTTC |
| KLHL40-KO-R | GTCTGTCCTAGCTTCCTCACTG |

## Radioactive in situ hybridization

Radioisotopic in situ hybridization (ISH) on E10.5, E12.5 and E15.5 embryonic sections was performed as previously described (*Shelton et al., 2000*). The following sequence was used to detect *Klhl41* expression:

AGCTGGAATCTAAAGAGTTTGCACCCACTGAAGTCAATGACATATGGAAGTATGAAGATGA
TAAAAAAGAATGGGCTGGGATGCTGAAGGAAATCCGTTACGCTTCGGGAGCTAGTTGCCTAG-
CAACGCGCTTAAATCTGTTTAAACTGTCTAAACTATAAAGGAGGTGACAAAGACACAGTTTGA-
GAGGTGGCTTGTTGGGACAAGAGGCTTTAATTTATTGTCATTCTTTAAGCCTATACAATGATTCACA
TAGGGTTACAGGGATTCACGCAGTTTCTCTGGGGTAAAACAGTGTAACCGAATCCCAGAGA
TTTTCAGTGTGCCAAGTATAAAACCATTTGCTAGGAAGTTTAGTATTCAGTTGAACAATATA
TTTTTTTTTTTTTTTTTTTTGGTTTTTTGAGACAGGGTTTCTCTGTATAGC

The DNA was cloned into PCRII-Topo vector (Life Technologies) as per manufacturer's instructions using the following primer set and skeletal muscle cDNA as a template:

Kl41-IS-F: 5'- cactgaagtcaatgacatatggaag-3',
Kl41-IS-R: 5'- gctatacagagaaaccctgtctcaa-3'.

Following cloning, probe sequence was cut out from the PCRII-Topo backbone and transcribed using the MAXIscript in vitro transcription kit (Life Technologies) by the T7 and SP6 promoters to generate the anti-sense and sense probes, respectively.

## Northern blot analysis

Tissues were harvested from 8 week old C57BL/6 mice and flash-frozen in liquid nitrogen. Total RNA was extracted from tissue with TRIZOL reagent (Invitrogen) according to manufacturer instructions. The in-situ hybridization probe sequence was used to generate the Northern blot probe by labeling with $[\alpha\text{-}^{32}P]dCTP$ using the RadPrime DNA Labelling System (Invitrogen), as per manufacturer's instructions. As a loading control, 28S and 18S rRNAs from gel run were visualized.

## RNA expression

RNA levels were measured by in situ hybridization, Northern blot analysis, RNA-seq, and qRT-PCR, as previously described (*Cenik et al., 2016*). RNA-seq (n = 3 mice per genotype) was performed by the UT Southwestern Genomic and Microarray Core Facility. For analysis of multiple groups, Holm-Sidak correction for multiple comparisons was utilized with a false discovery rate of 0.05. Data are available from Gene Expression Omnibus (GSE95543). For qRT-PCR, total RNA was extracted from tissue with TRIZOL reagent (Invitrogen) according to manufacturer instructions. cDNA was synthesized using Superscript III reverse transcriptase with random hexamer primers (Invitrogen), as per manufacturer instructions. Gene expression was analyzed by qRT-PCR using KAPA SYBR FAST (Kapa Biosystems). The following primers were used:

| Gene | Name | Sequence (5' to 3') |
| --- | --- | --- |
| *Klhl41* | RT1_Klhl41-E1.2-F | CTGTATGTGGACGAAGAAAATAAGG |
| *Klhl41* | RT1_Klhl41-E1.2-R | CCACCACATAGATTTTGTCATCTACT |
| *Klhl41* | RT1_Klhl41-E3.4-F | TTTTTCCAGCTTGATAACGTAACAT |
| *Klhl41* | RT1_Klhl41-E3.4-R | AGATTTTTCACTTCACTCCACTTTG |
| *Klhl41* | RT2_Klhl41-E3.4-F | CCAGCTTGATAACGTAACATCTGA |
| *Klhl41* | RT2_Klhl41-E3.4-R | AGATTTTTCACTTCACTCCACTTTG |
| *Klhl41* | RT1_Klhl41-E5.6-F | GAAGATGGTCTTTCAGCTTCAGTT |
| *Klhl41* | RT1_Klhl41-E5.6-R | AGTGGGTGCAAACTCTTTAGATTC |
| *Klhl40* | RT_Klhl40-F | CCCAAGAACCATGTCAGTCTGGTGAC |
| *Klhl40* | RT_Klhl40-R | TCAGAGTCCAAGTGGTCAAACTGCAG |
| *Lmod3* | RT_Lmod3-F | CCGCTGGTGGAAATCACTCCC |
| *Lmod3* | RT_Lmod3-R | ACTCCAGCTCCTTTGGCAGTTGC |
| *Neb* | RT_Neb-F | TGACTTGAGAAGTGATGCCATTC |
| *Neb* | RT_Neb-R | CTCTAGCGCCAATGTGGTGAC |

## Histology, immunochemistry and electron microscopy

Skeletal muscle tissues were flash-frozen in a cryoprotective 3:1 mixture of tissue-freezing medium (Triangle BioSciences International) and gum tragacanth (Sigma-Aldrich) as previously described, followed by sectioning on a cryostat (*Liu et al., 2014*). Hearts and diaphragms were fixed in 4% paraformaldehyde, followed by paraffin embedding and sectioning. Routine hematoxylin and eosin staining was performed on both paraffin-embedded tissue and cryosections as previously described (*Cenik et al., 2015*). Desmin (M0760, clone D33; Dako), laminin (L9393; Sigma-Aldrich) and sarcomeric α-actinin (A7811, Sigma-Aldrich) staining was performed on cryosections of skeletal muscle (*Millay et al., 2013*). Gomori's trichrome staining was performed as previously described (*Frey et al., 2004*). For electron microscopy, processing of muscle tissues was performed as previously described (*Nelson et al., 2013*). Images were acquired using a FEI Tecnai G2 Spirit Biotwin transmission electron microscope.

## Plasmid constructs

Tagged KLHL40, LMOD3 and NEB$_{frag}$ plasmids were cloned as previously described (*Garg et al., 2014*). A fragment of FLNC (amino acids 2133–2725) cloned into pcDNA3.1-FLAG was used from previous studies (*Frey et al., 2000*). A fragment of NRAP (NCBI reference sequence NM_008733.4, nucleotides 162–1106) was cloned into pcDNA3.1-myc by conventional PCR using a synthetic gBlock (Integrated DNA Technologies) as template. KLHL41 was cloned from mouse quadriceps cDNA using Phusion High-Fidelity DNA Polymerase (NEB) and the following primers:

    KLHL41-F: gatgaattcGATTCCCAGCGGGAGCTTGCAGA
    KLHL41-R: tgctcgagTTATAGTTTAGACAGTTTAAACAGATTTAAGCGCG

    KLHL41 was then subcloned into pcDNA3.1-FLAG, pcDNA3.1-myc and pCS2-3xFLAG-HA. For tandem affinity purification, N-terminal tagged pCS2-3xFLAG-HA-KLHL41 was subcloned into pBx vector. Domain deletions of KLHL41 were generated by conventional PCR and cloned into pcDNA3.1-FLAG. The deleted regions were (numbers correspond to nucleotides in NCBI reference sequence NM_028202.3): ΔBTB (78-467), ΔBACK (518-788) and ΔKR (789-1898). For generation of ΔBACK, 2 PCR products were used for triple ligation. Domain deletion mutants were subcloned into pCS2-3xFLAG-HA. NEB$_{frag}$ deletions were generated by conventional PCR and triple ligation into pcDNA3.1(+). pRK5-HA-Ub-WT and pRK5-HA-Ub-K0 were a gift from Ted Watson (Addgene plasmids #17608 and #17603 respectively) (*Lim et al., 2005*). The remaining HA-Ubiquitin mutants were cloned by conventional mutagenesis (QuikChange Lightning Site-Directed Mutagenesis Kit, Agilent) using HA-Ub-WT as template. CUL3 was cloned from mouse quadriceps cDNA into C-terminal myc pcDNA3.1 (Invitrogen) using Phusion High-Fidelity DNA Polymerase (NEB) and the following primers:

    CUL3-F: TAAGCAGGTACCCGCCACCATGTCGAATCTGAGCAAAGGC
    CUL3-R: TGCTTACTCGAGTGCTACATATGTGTATACTTTGCGATC

## Western blot analysis on skeletal muscle samples

Flash-frozen muscle tissues were dissociated in IP-A lysis buffer (50 mM Tris, 150 mM NaCl, 1 mM EDTA, 1% Triton, supplemented with cOmplete mini EDTA protease inhibitor cocktail (Sigma) and PhosSTOP phosphatase inhibitor cocktail (Sigma)) using a mini pestle and mechanical disruption with a 25G 5/8 needle. Tissue lysate was centrifuged at 20,817 x $g$ for 15 min at 4°C and the supernatant containing protein was transferred to a new tube. Protein was mixed with one volume of 2X Laemmli buffer with 5% β-mercaptoethanol and SDS-PAGE electrophoresis was performed following 5 min of sample incubation at 100°C. All protein was transferred to Immobilon-P PVDF membrane (EMD Millipore) using a Mini Trans-Blot cell (Bio-Rad). NEB western blot was performed as previously described (*Zhang et al., 2017*). All antibodies were diluted in 5% non-fat milk in TBS/0.1% Tween-20. Primary antibodies: anti-KLHL41 (ab66605 Abcam 1:2,000), anti-LMOD3 (14948–1-AP, Proteintech. 1:500), anti-NEB (19706–1-AP, Proteintech, 1:200) and anti-GAPDH (MAB374, EMD Millipore, 1:10,000). All primary antibodies were incubated with blots overnight at 4°C, while secondary antibodies were incubated for 1 hr at room temperature.

## Tandem affinity purification of KLHL41 from C2C12 cells and protein identification

Tandem affinity purification was performed as previously described (Garg et al., 2014). Briefly, 80% confluent Platinum E cells (Cell Biolabs) on 15 cm plates were transfected with pBx-3xFLAG-HA-GFP or pBx-3xFLAG-HA-KLHL41 using Fugene 6 (Promega) at a 3:1 DNA to Fugene 6 ratio according to manufacturer's directions. Viral media were collected 24, 36 and 48 hr after transfection and filtered through a 0.45 μm syringe filter (Corning). Polybrene was added to the media at a final concentration of 6 μg/mL. Media were added to 80% confluent C2C12 myoblasts (American Type Culture Collection) in growth media (DMEM with 10% fetal bovine serum and 1% antibiotic-antimycotic) (Life Technologies). C2C12 cells were then washed with PBS and differentiated for 5 days with differentiation media (DMEM with 2% horse serum and 1% antibiotic-antimycotic) (Life Technologies). Protein was collected from C2C12 myotubes by washing cells and then collecting them into 5 mL of PBS by scraping cells with a sterile cell lifter. Protein extraction was done in IP-A buffer according to standard procedures. Anti-FLAG M2 affinity gel (A2220, Sigma) and EZView Red anti-HA affinity gel (E6779, Sigma) were used for pull-down after equilibration in IP-A buffer. 3xFLAG peptide (F4799, Sigma) and HA peptide (I2149, Sigma) were used for elution in IP-A buffer as per manufacturer's instructions. IP-B buffer (50 mM Tris, 700 mM NaCl, 1 mM EDTA, 1% Triton) was used to wash beads before elution. Silver staining and peptide identification were performed as previously described (Garg et al., 2014). C2C12 and PE cells were tested for mycoplasma contamination with Universal Mycoplasma Detection Kit (30–1012K, ATCC) and they were negative for mycoplasma.

## Protein analysis in COS-7 cells

COS-7 cells (American Type Culture Collection) in 60 mm dishes were transfected at 75% confluency using Fugene 6 (Promega) as per manufacturer's directions. A total of 5 ug of DNA was used for each transfection. 48 hr after transfection, cells were washed with PBS and lysed in IP-A for 15 min. Lysates were rotated for 1 hr at 4°C and centrifuged at 20,817 x $g$ for 15 min. The pellet was discarded. Standard SDS-PAGE was performed. FLAG M2 (F1804, Sigma, 1:5,000), mouse c-myc (R950-25, ThermoFisher, 1:5,000), rabbit c-myc (sc-789, Santa Cruz, 1:500), HA (32–6700, ThermoFisher, 1:5,000), GAPDH (MAB374, EMD Millipore 1:20,000), LC3 (NB100-2220, Novus Biologicals, 1:2,000) and fibrillarin (ab5821, Abcam, 1:1,000) antibodies were used for blotting. For solubility assays, the remaining pellet was solubilized in high detergent lysis buffer (50 mM Tris, 150 mM NaCl, 1% NP-40, 0.1% SDS), boiled and incubated for 24 hr at 4°C under constant mixing.

For ubiquitination assays, cells were lysed in IP-A buffer supplemented with 10 mM N-ethylmaleimide (E3876, Sigma), a deubiquitinase inhibitor. For HA-Ub co-immunoprecipitation, cells were lysed in high detergent lysis buffer with 10 mM N-ethylmaleimide and the final volume was diluted 1:2 with IP-A buffer.

For chemical treatments, MG132 (C2211, Sigma) or chloroquine (C6628, Sigma) was dissolved in DMSO and added to COS-7 at a final concentration of 10 μM for 24 hr and 12 hr, respectively. For cycloheximide pulse experiments, cycloheximide (C7698, Sigma) was resuspended in ethanol and added to COS-7 cells at a final concentration of 100 μg/ml.

COS-7 cells were tested for mycoplasma contamination with Universal Mycoplasma Detection Kit (30–1012K, ATCC) and they were negative for mycoplasma.

## Immunofluorescence in COS-7 cells

COS-7 cells were plated in μ-Slide 8 Well ibiTreat (80826, ibidi) and transfected with the indicated plasmids. 30 hr later, cells were fixed in 4% paraformaldehyde for 15 min, permeabilized in 0.2% Triton X-100 for 20 min and blocked in 5% BSA for 1 hr. Primary antibody incubation was performed for 1 hr with antibodies against c-myc (R950-25, ThermoFisher), FLAG (F7425, Sigma) and fibrillarin (ab5821, Abcam). Primary and secondary Alexa-Fluor antibodies were used at 1:200 dilution.

## Quantitative proteomic analysis of skeletal muscle

Hindlimb muscle from P0 mice (n = 3 mice per genotype) was collected for quantitative proteomic analysis by 10-fraction LC/LC-MS/MS by Proteomics and Metabolomics Shared Resource at Duke University. For sample preparation, each sample was added 8 M urea in 50 mM ammonium bicarbonate, pH 8.0 at a constant 10 μL per mg of tissue. Samples were then subjected to mechanical

disruption using a Tissue Tearer followed by 2 rounds of probe sonication on ice at 30% power. Samples were spun to remove insoluble material and 4 μL was removed and subjected to Bradford assay to determine protein quantity. From each sample, 100 μg of total protein was removed and concentrations were normalized. Samples were then diluted in 1.6M urea with 50 mM ammonium bicarbonate. All samples were reduced for 20 min at 80°C with 10 mM dithiothreitol and alkylated for 40 min at room temperature with 25 mM iodoacetamide. Trypsin was added to a 1:50 ratio (enzyme to total protein) and allowed to proceed for 18 hr at 37°C. Samples were then acidified with 0.2% TFA (pH 2.5) and subjected to C18 SPE cleanup (Sep-Pak 50 mg bed). Following elution, all samples were frozen and lyophilized to dryness.

For TMT labeling, each sample was resuspended in 100 μL 200 mM triethylamonium bicarbonate, pH 8.0. Fresh TMT reagents (0.8 mg for each 6-plex reagent) were resuspended in 100 μL acetonitrile. 50 μL of each TMT tag was added to a specific sample and incubated for 4 hr at room temperature. Afterwards, 8 μL 5% hydroxylamine was added to quench the reaction. 20% of each sample were combined at 1:1:1:1:1:1 ratio and was then lyophilized to dryness prior to LC/LC-MS/MS analysis.

Quantitative two-dimensional liquid chromatography-tandem mass spectrometry (LC/LC-MS/MS) was performed on approximately 5 μg of protein digest per sample. The method uses two-dimensional liquid chromatography in a high-low pH reverse phase/reverse phase configuration on a nano-Acquity UPLC system (Waters Corp.) coupled to a Thermo QExactive Plus high resolution accurate mass tandem mass spectrometer with nanoelectrospray ionization. Peptides were first trapped at 2 ul/min at 97/3 v/v water/MeCN in 20 mM ammonium formate (pH 10) on a 5 μm XBridge BEH130 C18 300 μm x 50 mm column (Waters Corp.). A series of step-elutions of MeCN at 2 μl/min was used to elute peptides from the 1st dimension column. Ten steps of 14.0%, 16.0%, 17.3%, 18.5%, 20.3%, 22.0%, 23.5%, 25.0%, 30.0% and 50.0% MeCN were utilized for the analyses; these percentages were optimized for delivery of an approximately equal load to the 2nd dimension column for each fraction. For 2nd dimension separation, the elution of the 1st dimension was first diluted 10-fold online with 99.8/0.1/0.1 v/v/v water/MeCN/formic acid and trapped on a 5μ Symmetry C18 180 μm x 20 mm trapping column (Waters Corp.). The 2nd dimension separations were performed using a 1.7 μm Acquity BEH130 C18 75mmx250mm column (Waters Corp.) using a 90 min gradient of 3% to 25% acetonitrile with 0.1% formic acid at a flow rate of 400 nL/min with a column temperature of 55°C. Data collection on the QExactive Plus mass spectrometer was performed in a data-dependent acquisition mode of acquisition with a r = 70,000 (at m/z 200) full MS scan from m/z 375–1600 with a target AGC value of 1e6 ions followed by 20 MS/MS scans at r = 17,500 (at m/s 200) at a target AGC value of 5e4 ions. A 30 s dynamic exclusion was employed to increase depth of coverage. The total analysis cycle time for each sample injection was approximately 2 hr.

Following the 10 LC-MS/MS analyses, raw data were processed by Protein Discoverer to create MGF files. These MS/MS data were searched against a SwissProt_Mouse database within Mascot Server (Matrix Science) that also contained a reversed-sequence 'decoy' entry for each protein for false positive rate determination. Because mouse nebulin (NEB) is not a reviewed entry in SwisProt_Mouse, the unreviewed entry E9Q1W3_MOUSE was manually included in the analysis. Search tolerances were 5ppm precursor and 0.02 Da product ions with full trypsin protease rules and up to two missed cleavages. Search results were imported to Scaffold Q + S v4.4.6 (Proteome Software) and data was annotated at a Protein False Discovery Rate of 1.0%.

The overall dataset yielded identifications for 23,910 TMT labeled peptides corresponding to 4,418 TMT labeled proteins. Only peptides uniquely identified to a protein were considered. To normalize the six different channels to account for differences in labeling efficiencies and mixing percentages, the summed intensity for each channel was calculated and then normalized to the 126 channel. Then, protein level intensities were generated by summing all of the unique peptide intensities to that protein. A list of proteins differentially changed between WT and KO is presented in *Figure 4—source data 1*.

For representation, proteins more than 30% up- or down- regulated were selected as input for Morpheus (https://software.broadinstitute.org/morpheus/). Z-score was used for heat map scale. For pathway analysis, a list of significantly up- or down-regulated proteins was used as input for DAVID analysis of enriched GO Terms related to biological pathways (https://david.ncifcrf.gov) (*Jiao et al., 2012*).

## Statistical analysis

Data were presented as mean ±SEM. Differences between two groups were tested for statistical significance using the unpaired two-tailed Student's t test. $p < 0.05$ was considered significant. For analysis of multiple groups, we utilized the Holm-Sidak correction for multiple comparisons with a false discovery rate of 0.05.

## Acknowledgements

We thank Kate Luby-Phelps for assistance in the UT Southwestern Electron Microscopy Core Facility as well as John Shelton and James Richardson for help with histology. We thank Xiang Chen for technical assistance. We thank Erik Soderblom from Duke University School of Medicine for the use of the Proteomics and Metabolomics Shared Resource, which provided the quantitative proteomics service. We thank Jose Cabrera for assistance with graphics. This work was supported by grants from the NIH (HL130253, HL-077439, DK-099653 and AR-067294), Senator Paul D. Wellstone Muscular Dystrophy Cooperative Research Center grant (U54 HD 087351) and the Robert A Welch Foundation (grant 1–0025 to ENO).

## Additional information

### Funding

| Funder | Grant reference number | Author |
| --- | --- | --- |
| National Institutes of Health | HL130253 | Eric N Olson |
| National Institutes of Health | HL077439 | Eric N Olson |
| National Institutes of Health | DK099653 | Eric N Olson |
| National Institutes of Health | AR067294 | Eric N Olson |
| Welch Foundation | 1-0025 | Eric N Olson |

The funders had no role in study design, data collection and interpretation, or the decision to submit the work for publication.

### Author contributions

Andres Ramirez-Martinez, Conceptualization, Data curation, Formal analysis, Validation, Investigation, Writing—original draft, Writing—review and editing; Bercin Kutluk Cenik, Data curation, Formal analysis, Visualization, Methodology; Svetlana Bezprozvannaya, Data curation, Methodology; Beibei Chen, Data curation, Formal analysis; Rhonda Bassel-Duby, Conceptualization, Formal analysis, Supervision, Funding acquisition, Investigation, Writing—original draft, Project administration, Writing—review and editing; Ning Liu, Conceptualization, Data curation, Formal analysis, Supervision, Validation, Investigation, Visualization, Writing—original draft, Project administration, Writing—review and editing; Eric N Olson, Conceptualization, Resources, Formal analysis, Supervision, Funding acquisition, Validation, Visualization, Writing—original draft, Project administration, Writing—review and editing

### Author ORCIDs

Eric N Olson [ID] http://orcid.org/0000-0003-1151-8262

### Ethics

Animal experimentation: This study was performed in strict accordance with the recommendations in the Guide for the Care and Use of Laboratory Animals of the National Institutes of Health. All of the animals were handled according to approved institutional animal care and use committee (IACUC) protocols (2015-100829) of the University of Texas Southwestern Medical Center. The protocol was approved by the Committee on the Ethics of Animal Experiments of the University of Texas Southwestern Medical Center (NIH OLAW Assurance Number D16-00296 ).

Decision letter and Author response
Decision letter https://doi.org/10.7554/eLife.26439.023
Author response https://doi.org/10.7554/eLife.26439.024

## Additional files
### Supplementary files
• Transparent reporting form
DOI: https://doi.org/10.7554/eLife.26439.020

### Major datasets
The following dataset was generated:

| Author(s) | Year | Dataset title | Dataset URL | Database, license, and accessibility information |
|---|---|---|---|---|
| Ramirez-Martinez A, Olson EN | 2017 | KLHL41 stabilizes skeletal muscle sarcomeres by nonproteolytic ubiquitination | https://www.ncbi.nlm.nih.gov/geo/query/acc.cgi?acc=GSE95543 | Publicly available at the NCBI Gene Expression Omnibus (accession no: GSE95543) |

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
