## [Decision Letter]

Thank you for submitting your article "KLHL41 stabilizes skeletal muscle sarcomeres by nonproteolytic ubiquitination" for consideration by *eLife*. Your article has been reviewed by three peer reviewers, one of whom, Amy Wagers is a member of our Board of Reviewing Editors and the evaluation has been overseen by Amy Wagers and Didier Stainier as the Senior Editor. The following individual involved in review of your submission has agreed to reveal their identity: Alan Beggs (Reviewer #2).

The reviewers have discussed the reviews with one another and the Reviewing Editor has drafted this decision to help you prepare a revised submission.

Summary:

Nemaline myopathy (NM) is congenital disease that effects sarcomere function and muscle defects. Several disease alleles have been mapped for NM, and these include two Kelch domain containing proteins that are likely substrate adaptors for the Cul3-based Cullin Ring E3 ubiquitin ligase. The molecular contribution of KLHL41 to NM disease has not yet been determined.

This manuscript describes the generation and analysis of KLHL41-deficient mice, in which expression and function of KLHL41 is ablated. The authors show that KLHL41 knockout mice develop a severe and fatal myopathy that resembles human NM, and that this phenotype is associated with a failure to stabilize Nebulin, and likely other proteins involved in maintaining sarcomere integrity, in the KO muscle cells. They further provide evidence that the normal role of KLHL41 in stabilizing Nebulin depends upon polyubiquitination of a particular lysine residue in the BTB domain of the protein, and suggest a somewhat unexpected chaperone function for the protein in this context. Finally, they draw distinctions in the functions of KLHL41 with respect to its close homolog KLHL40 (also mutated in patients with NM), arguing that these two Kelch protein possess both common and distinct targets for their actions (with 41 targeting only Nebulin, and 40 targeting both Nebulin and LMOD3 for stabilization).

Overall, the study is logically designed, carefully executed and thoughtfully presented. Most conclusions are well justified by the data presented and provide an important advance in our understanding of the biology of kelch proteins in normal and diseased muscle. However, the reviewers did raise concerns about two aspects of the authors model: (1) a failure to consider or exclude an alternative model in which KLHL41 ubiquitinates and degrades a third protein, which would normally function to ubiquitinate and degrade Nebulin, and (2) a lack of strong evidence indicating that KLHL41 does not interact with LMOD3. These concerns are further articulated below, along with suggestions for experiments to address them and for improving the clarity and transparency of the presentation.

Essential revisions:

1) The most significant concern with this study is that the authors fail to rule out a very simple, alternative hypothesis. That is, that KLHL41 is an E3 ubiquitin ligase for a second E3 enzyme that promotes the degradation of NEB. Thus, when KLHL41 is overexpressed, this leads to the ubiquitin mediated, k48-chain dependent, degradation of another E3, and that in the absence of this second enzyme NEB accumulates to high levels. Notably, there are many examples of E3 ligase cross-talk in the ubiquitin system (e.g. the cell cycle E3 APC/C controls the degradation of SKP2, an adaptor for the SCF). The authors need to address this by identifying the binding site (degron) between KLHL41 and NEB that controls the regulation. When that degron site is mutated the regulation should be lost, akin to mutating a kinase consensus motif thereby blocking signaling. As an alternative, they might show that in vitro KLHL41 can ubiquitylate NEB, which would be supportive of the direct regulation they propose, but not as convincing as the degron experiment. As the authors' conclusion that K48 linked ubiquitination is driving protein stability and not degradation through the proteasome is highly unusual, and there are few if any concrete examples of this happening, the data supporting this conclusion should be very strong.

2) There is a correlation between KLHL41 being ubiquitylated and NEB being stabilized.

However, there is little reason to directly connect these two phenomena. Many, if not most, CRL substrate receptors are ubiquitylated by their respective E3s. KLHL41 could be ubiquitylated, and NEB degradation could be dependent on K48 ubiquitylation, but these two things might have nothing to do with each-other. As stated above, a much more simple hypothesis is that KLHL41 targets an NEB ligase for degradation.

3) The authors' conclusion that KLHL41 fails to stabilize LMOD3, which represents the sole evidence presented for a function distinct from that of KLHL40 is somewhat unconvincing. It appears from the blot in Figure 4 that KLHL41 does stabilize LMOD3 to some degree, although clearly not to the same level as KLHL40. Adding to the concern, while the authors state that they did not see co-immunoprecipation of KLHL41 with LMOD3, this data is not actually included in Figure 4. It would also have been helpful if they had performed some sort of immunodetection (e.g. Western) of the TAP-isolated proteins (Figure 4) to confirm the lack of LMOD3 association in this study. Anyway, since the distinctive interactions of KLHL41 is presented as a major conclusion of the paper, and depends almost entirely on whether or not stabilization of LMOD3 occurs, the authors should provide additional evidence to strengthen these data, or temper their conclusions. Finally, though perhaps not necessary for the current manuscript, an experiment in which the KLHL40 and KLHL41 domains suspected by the authors as mediating their distinct activities are swapped and analyzed for changes in NEB/LMOD3 stabilization capacity would add significantly to the study.

4) Figure 4: What criteria were applied to select the proteins for display in the heat map? Was there are fold-change threshold or other quantitative criterion? The statement is made that "A remarkable number of up-regulated proteins were involved in ubiquitination,…" (subsection “KLHL41 is required to maintain normal levels of sarcomeric proteins in vivo”). Have, or can, the authors apply a statistical method such as DAVID or other form of Gene Set Enrichment Analysis to support this, and are there other functional groups of proteins exhibiting unusual degrees of up or down regulation?

5) The authors state in subsection “KLHL41 is required to maintain normal levels of sarcomeric proteins in vivo” that NEB is the second most down-regulated protein in KLHL41 KO muscle, but it appears to be farther down the list in their excel file? What about the 12 other proteins (matn2, matn1, lgmn, etc)?

6) Figure 1 – data and experimental details presented are insufficient for the conclusions drawn and there is no description of these studies in the methods section. How many different litters are represented in the table? How many different sets of parents? Data also lack statistical assessment evaluating divergence from expected values, and specific phenotypes observed/assayed in mice (e.g. premature death, runting, etc.) of each genotype should be included in the table together with the genotype.

7) It is not clear what the frequency and distribution of sarcomeric disarray, Z line streaming, and electron dense inclusions is in the examined tissues in Figure 3 and the supplement Figure 3. Each of these findings may often be seen in normal muscles. The authors should provide some semi-quantitative estimation or more detailed description of their nature and abundance in KLHL41 KO muscles relative to WT.

8) NEB transcript levels should be confirmed by qPCR. While NEB transcripts were found in the RNA seq data, the conclusions here hinge on the existence of post-translational mechanism. It says this data is in Figure 4—figure supplement 1, but only lmod3 is checked there.

9) An immunoblot of NEB in muscles from control and KO mice is never shown. This is essential as it would confirm that NEB is in fact downregulated after KLHL41 KO.

10) Consistent with their predictions and mass spec data, the authors show in many experiments that KLHL41 overexpression increases the levels of NEB. It is essential that they also show that KLHL41 depletion reduces NEB levels, and this should be done using multiple RNAi reagents (or crispr KO).

11) The authors should clarify how Figure 7 and Figure 7 reveal a role for Klhl41 in shifting the pool of NEB from the insoluble pellet to the soluble fraction. How is this distinguished from the stabilization of NEB seen in all of the previous figures? For the IF in Figure 7, this would require an enormous amount of normalization since the total fluorescence should be significantly higher in the presence of KLHL41. If this was done, how was it achieved?

[Editors' note: further revisions were requested prior to acceptance, as described below.]

Thank you for resubmitting your work entitled "KLHL41 stabilizes skeletal muscle sarcomeres by nonproteolytic ubiquitination" for further consideration at *eLife*, and sorry for the delay in the reviewing process. Your revised article has been favorably evaluated by Didier Stainier (Senior editor), a Reviewing editor (Amy Wagers), and two reviewers.

The manuscript has been provisionally accepted, but there are a few remaining issues that need to be addressed before final acceptance and publication, as outlined below. All of these issues should be addressable by simple revision to the text.

1) The authors demonstrate clearly that KLHL41 expression positively modulates the abundance of NEB. Mechanistically, the argue that KLHL41 gets ubiquitylated (which they show clearly) and that this in turns increases its ability to rescue NEB for an insoluble material, suggesting it acts as a chaperone and this is dependent on its k48-linked ubiquitylation. While the data demonstrating that KLHL41 enhances NEB levels, preventing its aggregation and that this depends on ubiquitin chain formation is compelling, the current manuscript falls short of showing that this is specifically dependent on KLHL41 ubiquitylation. Rather than showing that the "stabilizing activity of KLHL41 is dependent on K48-linked poly-ubiquitination of KLHL41." (quoted from the abstract), the data actually show that KLHL41 is ubiquitylated on K48 AND that KLHL41 stabilizes NEB. This is an important distinction, and, particularly as the authors are proposing a largely new mechanism, they should tone down the language they use (throughout the paper) to make this mechanistic argument.

2) In Subsection “KLHL41 and KLHL40 stabilize different substrates despite their high homology”, the authors state that "These results indicate that KLHL41 stabilizes NEB_frag_ but not LMOD3, confirming the functional distinctions between KLHL40 and KLHL41."; however, as discussed in the prior review and apparent from the data in Figure 4, KLHL41 stabilizes both NEB and LMOD3, although its stabilizing activity for LMOD3 is not as robust as that of KLHL40. Similarly, in subsection “KLHL40 and KLHL41 stabilize their partners via distinct mechanisms”: "only KLHL40 stabilizes LMOD3 significantly" – it is unclear from the data that the modest stabilization of LMOD3 by KLHL41 is insignificant. These sentences should be revised to more accurately represent the data.

3) Figure 6 – HA-Ub-KO seems to be labeled KO instead of K0 (zero), which could be confusing. It is labeled correctly in the other panels (B and C). Also, it does not appear that Ub-K0 is overexpressed in the absence of KLHL41 in this experiment, in contrast to what is stated in the text (subsection “KLHL41 stabilizing activity is regulated through K48-linked poly-ubiquitination of the BTB domain”).

4) Change KO to K0 in Figure 6—figure supplement 1 also.

5) Subsection “KLHL41 prevents aggregation of NEB_frag_ “states that "all mutants exhibited increased protein levels in the presence of KLHL41…", and references Figure 7; however, I do not see such data in this figure. The top blot does not evaluate protein abundance of the deletion mutants in the presence of KLHL41, and the bottom blot is an IP, which cannot separate changes in protein stability from differences in association with KLHL41. Please revise the text or add the relevant data.

---

## [Author Response]

*Essential revisions:*

*1) The most significant concern with this study is that the authors fail to rule out a very simple, alternative hypothesis. That is, that KLHL41 is an E3 ubiquitin ligase for a second E3 enzyme that promotes the degradation of NEB. Thus, when KLHL41 is overexpressed, this leads to the ubiquitin mediated, k48-chain dependent, degradation of another E3, and that in the absence of this second enzyme NEB accumulates to high levels. Notably, there are many examples of E3 ligase cross-talk in the ubiquitin system (e.g. the cell cycle E3 APC/C controls the degradation of SKP2, an adaptor for the SCF). The authors need to address this by identifying the binding site (degron) between KLHL41 and NEB that controls the regulation. When that degron site is mutated the regulation should be lost, akin to mutating a kinase consensus motif thereby blocking signaling. As an alternative, they might show that* in vitro *KLHL41 can ubiquitylate NEB, which would be supportive of the direct regulation they propose, but not as convincing as the degron experiment. As the authors' conclusion that K48 linked ubiquitination is driving protein stability and not degradation through the proteasome is highly unusual, and there are few if any concrete examples of this happening, the data supporting this conclusion should be very strong.*

We agree with the reviewer that a second E3 enzyme could be degrading NEB. However, our data argues against the existence of another E3 enzyme. If a second E3 enzyme was being degraded by KLHL41, inhibition of protein degradation would be sufficient to stabilize NEB_frag_ even in the absence of KLHL41. However, NEB_frag_ failed to accumulate by addition of MG132 or overexpression of Ubiquitin(K0), which precludes the involvement of another E3 enzyme (Figure 7 and Figure 6).

Furthermore, in an attempt to identify a potential degron in NEB_frag_, we generated NEB_frag_ mutants with small deletions spanning the whole protein and tested their stability with and without KLHL41 (revised Figure 7 and revised Figure 1—figure supplement 1). Our results show most mutants were stable in the absence of KLHL41, indicating that there is no degron being targeted by an unknown E3 ubiquitin ligase. Moreover, the fact that most deletion mutants are stable indicates that KLHL41 stabilizes NEB by a nonproteolytic mechanism.

*2) There is a correlation between KLHL41 being ubiquitylated and NEB being stabilized.*

*However, there is little reason to directly connect these two phenomena. Many, if not most, CRL substrate receptors are ubiquitylated by their respective E3s. KLHL41 could be ubiquitylated, and NEB degradation could be dependent on K48 ubiquitylation, but these two things might have nothing to do with each-other. As stated above, a much more simple hypothesis is that KLHL41 targets an NEB ligase for degradation.*

We agree these data may seem correlative but we have data to show the connection. As seen in Figure 6, deletion of the BTB of KLHL41 (ΔBTB) is able to rescue NEB levels in the presence of Ubiquitin(K0) while full length KLHL41 cannot. This observation the BTB domain. We highlighted this observation in the revised Results.

As mentioned in Point 1, we addressed the possibility of a NEB ligase by using NEB deletion mutants (revised Figure 7), and observed that most deletions rendered the protein stable in the absence of KLHL41. These findings, in addition to the results showing NEB_frag_ aggregation in the absence of KLHL41 (Figure 7) support KLHL41 acting as a chaperone. These points were included in the revised Discussion section.

*3) The authors' conclusion that KLHL41 fails to stabilize LMOD3, which represents the sole evidence presented for a function distinct from that of KLHL40 is somewhat unconvincing. It appears from the blot in Figure 4 that KLHL41 does stabilize LMOD3 to some degree, although clearly not to the same level as KLHL40. Adding to the concern, while the authors state that they did not see co-immunoprecipation of KLHL41 with LMOD3, this data is not actually included in Figure 4. It would also have been helpful if they had performed some sort of immunodetection (e.g. Western) of the TAP-isolated proteins (Figure 4) to confirm the lack of LMOD3 association in this study. Anyway, since the distinctive interactions of KLHL41 is presented as a major conclusion of the paper, and depends almost entirely on whether or not stabilization of LMOD3 occurs, the authors should provide additional evidence to strengthen these data, or temper their conclusions. Finally, though perhaps not necessary for the current manuscript, an experiment in which the KLHL40 and KLHL41 domains suspected by the authors as mediating their distinct activities are swapped and analyzed for changes in NEB/LMOD3 stabilization capacity would add significantly to the study.*

We agree that KLHL41 slightly increases LMOD3 levels (Figure 4). However, as the reviewer observed, this increase is not to the same extent as seen with KLHL40. We further validated the stabilization of LMOD3 by KLHL41 in vivo by western blot analysis of WT and KO muscle for LMOD3 (revised Figure 2). We observed a slight decrease in LMOD3 levels in KLHL41 KO, but not as dramatic as in KLHL40 KO, which shows an almost complete loss of LMOD3. We think that KLHL41 may stabilize LMOD3 at lower efficiency than KLHL40. We modified the manuscript to reflect these considerations.

We modified and optimized our co-immunoprecipitation experiments but were still unable to detect interaction between LMOD3 and KLHL41 in vitro (revised Figure 4—figure supplement 3). As previously reported (Garg et al., 2014), LMOD3 protein levels can be partially rescued in vitro by proteasome inhibition whereas NEB_frag_ is still unstable after treatment, which suggests that the stabilization of these proteins may occur through different pathways (protection from degradation versus misfolding). The lack of a direct interaction between LMOD3 and KLHL41 and the sensitivity of LMOD3 protein levels to proteasome inhibition indicate that in this case, LMOD3 stabilization could occur, at least partially, by degradation of another E3 ligase.

We analyzed KLHL40-KLHL41 chimeras as suggested by the reviewers. However, some constructs were not as stable as the original proteins, so we are not including these data in the revised manuscript.

*4) Figure 4: What criteria were applied to select the proteins for display in the heat map? Was there are fold-change threshold or other quantitative criterion? The statement is made that "A remarkable number of up-regulated proteins were involved in ubiquitination,…" (subsection “KLHL41 is required to maintain normal levels of sarcomeric proteins* in vivo*”). Have, or can, the authors apply a statistical method such as DAVID or other form of Gene Set Enrichment Analysis to support this, and are there other functional groups of proteins exhibiting unusual degrees of up or down regulation?*

We apologize for the confusion. For heat map display, we selected an arbitrary threshold of ± 1.3 fold change. In the revised manuscript, we also rearranged the heat map using Morpheus (https://software.broadinstitute.org/morpheus/) and expanded the figure legends for easier comprehension (revised Figure 4).

We performed DAVID analysis on the differentially expressed proteins (revised Figure 4—figure supplement 1). Among others, we identified “sarcomere organization” and “regulation of muscle contraction” as enriched pathways in the down-regulated proteins, whereas “protein ubiquitination involved in ubiquitin-dependent catabolic process” and “response to unfolded protein” were enriched in the up-regulated proteins. As indicated in the revised Results section, we referenced the DAVID analysis when discussing functional groups of proteins being regulated.

*5) The authors state in subsection “KLHL41 is required to maintain normal levels of sarcomeric proteins* in vivo*” that NEB is the second most down-regulated protein in KLHL41 KO muscle, but it appears to be farther down the list in their excel file? What about the 12 other proteins (matn2, matn1, lgmn, etc)?*

We apologize for the confusion. We rearranged the Excel file from most down-regulated (KLHL41) to most up-regulated (HIST1H1D) and modified the heat map (revised Figure 4) with Morpheus (https://software.broadinstitute.org/morpheus/).

We decided to focus on the regulation of NEB by KLHL41 because mutations in NEB are reported to be the main cause of nemaline myopathy. For that reason, we have not examined the role of the proteins indicated by the reviewer but agree that they may be interesting for future studies.

*6) Figure 1 – data and experimental details presented are insufficient for the conclusions drawn and there is no description of these studies in the methods section. How many different litters are represented in the table? How many different sets of parents? Data also lack statistical assessment evaluating divergence from expected values, and specific phenotypes observed/assayed in mice (e.g. premature death, runting, etc.) of each genotype should be included in the table together with the genotype.*

We did not document phenotypic or histological observations other than premature death. Since these data are not essential to our conclusions and may be more appropriate for another study, we removed these data from the manuscript.

*7) It is not clear what the frequency and distribution of sarcomeric disarray, Z line streaming, and electron dense inclusions is in the examined tissues in Figure 3 and the supplement Figure 3. Each of these findings may often be seen in normal muscles. The authors should provide some semi-quantitative estimation or more detailed description of their nature and abundance in KLHL41 KO muscles relative to WT.*

We quantified the number of protein aggregates (nemaline bodies) in WT and KO for a semi-quantitative assessment of abnormalities in muscle ultrastructure and included these data in revised Figure 3.

*8) NEB transcript levels should be confirmed by qPCR. While NEB transcripts were found in the RNA seq data, the conclusions here hinge on the existence of post-translational mechanism. It says this data is in Figure 4—figure supplement 1, but only lmod3 is checked there.*

We included data comparing Neb mRNA levels in WT and KO by qRT-PCR (revised Figure 4—figure supplement 2). No significant differences were found.

*9) An immunoblot of NEB in muscles from control and KO mice is never shown. This is essential as it would confirm that NEB is in fact downregulated after KLHL41 KO.*

In addition to proteomics data, we performed western blot analysis from WT and KO mice. We observed a reduction in NEB levels in KO mice (revised Figure 2). This and the previous results confirm that NEB is a target of KLHL41 and it is downregulated post-translationally in *Klhl41* KO mice.

*10) Consistent with their predictions and mass spec data, the authors show in many experiments that KLHL41 overexpression increases the levels of NEB. It is essential that they also show that KLHL41 depletion reduces NEB levels, and this should be done using multiple RNAi reagents (or crispr KO).*

We performed western blot analysis for NEB from WT and KO mice. Since we observed a reduction in NEB levels in KO mice (revised Figure 2), we have shown that KLHL41 depletion reduces NEB levels making it redundant to perform RNAi knockdown or CRISPR KO in cell lines.

*11) The authors should clarify how Figure 7 and Figure 7 reveal a role for Klhl41 in shifting the pool of NEB from the insoluble pellet to the soluble fraction. How is this distinguished from the stabilization of NEB seen in all of the previous figures? For the IF in 7D, this would require an enormous amount of normalization since the total fluorescence should be significantly higher in the presence of KLHL41. If this was done, how was it achieved?*

We apologize for any confusion. In our previous study of KLHL40 (Garg et al., 2014), we only examined NEB_frag_ protein levels in the soluble fraction. Because we could only detect NEB_frag_ in the presence of KLHL40, we concluded that NEB_frag_ protein was stabilized by an unknown mechanism (either protection from degradation, chaperone activity or both). In revised Figure 7 (formerly Figure 7), we detected NEB_frag_ in the insoluble pellet, which correlates with the nuclear aggregates observed by immunofluorescence in revised Figure 7 (formerly Figure 7). In the presence of KLHL41, these aggregates disappear and a fraction of NEB_frag_ shifts to the soluble fraction. Based on these observations, we concluded that KLHL41 stabilizes NEB_frag_ because it prevents NEB_frag_ aggregation, which makes it more soluble and detectable in the soluble fraction.

Regarding normalization and differences between immunofluorescence and western blots, the total levels of NEB_frag_ are higher in both experiments in the presence of KLHL41 compared to NEB_frag_ alone. To assess total levels of NEB_frag_ by western blot analysis, the intensities from soluble and insoluble fractions must be added, as the pellet was solubilized after the soluble fraction was obtained, which reflects the changes also observed by immunofluorescence. We modified the text to clarify this point in the Figure legends (revised Figure 7).

[Editors' note: further revisions were requested prior to acceptance, as described below.]

*The manuscript has been provisionally accepted, but there are a few remaining issues that need to be addressed before final acceptance and publication, as outlined below. All of these issues should be addressable by simple revision to the text.*

*1) The authors demonstrate clearly that KLHL41 expression positively modulates the abundance of NEB. Mechanistically, the argue that KLHL41 gets ubiquitylated (which they show clearly) and that this in turns increases its ability to rescue NEB for an insoluble material, suggesting it acts as a chaperone and this is dependent on its k48-linked ubiquitylation. While the data demonstrating that KLHL41 enhances NEB levels, preventing its aggregation and that this depends on ubiquitin chain formation is compelling, the current manuscript falls short of showing that this is specifically dependent on KLHL41 ubiquitylation. Rather than showing that the "stabilizing activity of KLHL41 is dependent on K48-linked poly-ubiquitination of KLHL41." (quoted from the abstract), the data actually show that KLHL41 is ubiquitylated on K48 AND that KLHL41 stabilizes NEB. This is an important distinction, and, particularly as the authors are proposing a largely new mechanism, they should tone down the language they use (throughout the paper) to make this mechanistic argument.*

We agree with the distinction and modified the manuscript to better represent our mechanistic data. While we have shown that KLHL41 is poly-ubiquitinated at the BTB domain (Figure 6) and that poly-ubiquitination inhibition prevents KLHL41 activity (Figure 6) in a BTB-dependent manner (Figure 6), the evidence for the role K48-linked ubiquitination of KLHL41 (Figure 6) requires further work. To reflect these considerations, we have modified the manuscript: Abstract, Introduction, subsection “KLHL41 stabilizing activity is regulated through K48-linked poly-ubiquitination of the BTB domain “, Discussion and legend for Figure 6.

*2) in Subsection “KLHL41 and KLHL40 stabilize different substrates despite their high homology”, the authors state that "These results indicate that KLHL41 stabilizes NEBfrag but not LMOD3, confirming the functional distinctions between KLHL40 and KLHL41."; however, as discussed in the prior review and apparent from the data in Figure 4, KLHL41 stabilizes both NEB and LMOD3, although its stabilizing activity for LMOD3 is not as robust as that of KLHL40. Similarly, in subsection “KLHL40 and KLHL41 stabilize their partners via distinct mechanisms”: "only KLHL40 stabilizes LMOD3 significantly" – it is unclear from the data that the modest stabilization of LMOD3 by KLHL41 is insignificant. These sentences should be revised to more accurately represent the data.*

We agree with the observation. We modified both statements to indicate that LMOD3 is stabilized by KLHL41 albeit at lower levels than by KLHL40 (subsection “KLHL41 and KLHL40 stabilize different substrates despite their high homology “and subsection “KLHL40 and KLHL41 stabilize their partners via distinct mechanisms”) and legend for Figure 4.

*3) Figure 6 – HA-Ub-KO seems to be labeled KO instead of K0 (zero), which could be confusing. It is labeled correctly in the other panels (B and C). Also, it does not appear that Ub-K0 is overexpressed in the absence of KLHL41 in this experiment, in contrast to what is stated in the text (subsection “KLHL41 stabilizing activity is regulated through K48-linked poly-ubiquitination of the BTB domain”).*

We corrected the labels for Figure 6 and Figure 6—figure supplement 1. We deleted the statement about Ub-K0 overexpression in the absence of KLHL41 for clarification (subsection “KLHL41 stabilizing activity is regulated through K48-linked poly-ubiquitination of the BTB domain”).

*4) Change KO to K0 in Figure 6—figure supplement 1 also.*

We corrected the labels for Figure 6 and Figure 6—figure supplement 1.

*5) Subsection “KLHL41 prevents aggregation of NEB_frag_ “states that "all mutants exhibited increased protein levels in the presence of KLHL41…", and references Figure 7; however, I do not see such data in this figure. The top blot does not evaluate protein abundance of the deletion mutants in the presence of KLHL41, and the bottom blot is an IP, which cannot separate changes in protein stability from differences in association with KLHL41. Please revise the text or add the relevant data.*

We agree we did not analyze NEB deletion mutants in the presence or absence of KLHL41 in the same western blot gel. However, both experiments (Figure 7 up and bottom) include full length NEBfrag+KLHL41 co-expression. We revised the text and compared each experiment to that control (subsection “KLHL41 prevents aggregation of NEBfrag “).